# TamGen: drug design with target-aware molecule generation through a chemical language model

Kehan Wu[1,5], Yingce Xia [2,5] ✉, Pan Deng [2,5], Renhe Liu[3,5], Yuan Zhang[3], Han Guo[3], Yumeng Cui[3], Qizhi Pei[4], Lijun Wu[2], Shufang Xie[2], Si Chen[3], Xi Lu[3], Song Hu[3], Jinzhi Wu[3], Chi-Kin Chan[3], Shawn Chen[3], Liangliang Zhou[3], Nenghai Yu[1], Enhong Chen[1], Haiguang Liu[2], Jinjiang Guo [3] ✉, Tao Qin [2] ✉ & Tie-Yan Liu [2]

Generative drug design facilitates the creation of compounds effective against pathogenic target proteins. This opens up the potential to discover novel compounds within the vast chemical space and fosters the development of innovative therapeutic strategies. However, the practicality of generated molecules is often limited, as many designs focus on a narrow set of drug-related properties, failing to improve the success rate of subsequent drug discovery process. To overcome these challenges, we develop TamGen, a method that employs a GPT-like chemical language model and enables target-aware molecule generation and compound refinement. We demonstrate that the compounds generated by TamGen have improved molecular quality and viability. Additionally, we have integrated TamGen into a drug discovery pipeline and identified 14 compounds showing compelling inhibitory activity against the Tuberculosis ClpP protease, with the most effective compound exhibiting a half maximal inhibitory concentration (IC$_{50}$) of 1.9 μM. Our findings underscore the practical potential and real-world applicability of generative drug design approaches, paving the way for future advancements in the field.

Generative drug design, a promising avenue for drug discovery, aims to create novel molecules/compounds with desired pharmacological properties from scratch, without relying on existing templates or molecular frameworks[1,2]. While conventional screening-based approaches, such as high-throughput screening, virtual screening, and emerging deep learning-based screening[3–6] usually hunt for drug candidates from libraries with $10^4$ to $10^8$ molecules[7–9], generative drug design enables exploration of the vast chemical space, which is estimated to contain over $10^{60}$ feasible compounds[10]. Consequently, it holds potential to identify underexplored classes of compounds, and

novel compounds that are not in any existing library. This is especially important for target proteins without hit compounds (starting point for drug design) and those having developed resistance to current drugs.

Generative modeling techniques greatly empowers drug design. In recent years, a growing number of approaches have been proposed to guide the generation of drug-like compounds given the information of target proteins, stemming from creative artificial intelligence techniques such as autoregressive models[11–15], generative adversarial networks (GAN)[16], variational autoencoders (VAE)[17,18], and diffusion

[1]University of Science and Technology of China, Hefei, China. [2]Microsoft Research AI for Science, Beijing, China. [3]Global Health Drug Discovery Institute, Beijing, China. [4]Renmin University of China, Beijing, China. [5]These authors contributed equally: Kehan Wu, Yingce Xia, Pan Deng, Renhe Liu. ✉e-mail: yingce.xia@microsoft.com; jinjiang.guo@ghddi.org; taoqin@microsoft.com

models[19–22]. These approaches, by exploring the chemical space conditioned on the target of interest, have demonstrated the feasibility of target-based generative drug design with deep learning. However, validations with biophysical or biochemical assays are often missing[23], as most of the generated compounds lack satisfying physiochemical properties for drug-like compounds such as synthetic accessibility. In other words, despite generating a large number of novel compounds, existing approaches struggle to demonstrate their capability to provide effective candidates that can improve the real-world drug discovery effectiveness.

We therefore propose a method named TamGen (Target-aware molecular generation). TamGen features a GPT-like chemical language model aiming for drug-like compound generation, inspired by the success of large language models[24]. The Generative Pre-trained Transformer[25] (GPT), backbone of large language models, has demonstrated its effectiveness in generating not only text[24] but also images[26] and speech[27], as well as understanding and solving scientific problems[28]. Here, we demonstrate that a GPT-like architecture and training strategy are also effective for generating chemical compounds in 1D space, as these compounds can be represented using Simplified Molecular Input Line Entry System (SMILES)[29], a sequential representation akin to text. In addition, we introduce two modules to encode target protein and compound information, which allow target-aware generation of compounds based on protein structures and compound refinement based on seeding compounds, respectively. With benchmark test, we show that TamGen not only produces compounds with higher plausibility, but also enhances the balance between pharmacological activity and synthetic accessibility.

We applied TamGen to generate compounds against tuberculosis (TB), an infectious disease caused by *Mycobacterium tuberculosis* (Mtb). TB was responsible for 1.3 million fatalities and 10.6 million new cases in 2022[30,31], and the rising antimicrobial resistance (AMR) in tuberculosis necessitates urgent therapeutic innovation to tackle the disease[32,33]. We focused on Caseinolytic protease P (ClpP), an essential serine protease in bacterial protein degradation system and an emerging novel target for antibiotic development[34–37]. Using a Design-Refine-Test pipeline powered by TamGen, we discovered 14 candidate compounds showing promising potency against Mtb ClpP, with half maximal inhibitory concentrations (IC$_{50}$) ranging from 1.88 μM to 35.2 μM. Significantly, the compounds generated by TamGen not only enrich candidate pool for further optimization, but also provide effective anchors for hit expansion and structure-activity relationship (SAR) synthesis. These findings highlight the broad applicability and considerable potential of TamGen in target-aware drug design.

## Results

### TamGen enables target-aware compound design and refinement
We implemented TamGen with three modules: (1) compound decoder, a GPT-like chemical language model and the core component of TamGen, which lays the foundation for compound generation in chemical space; (2) protein encoder, a Transformer-based model used to encode the binding pockets of target proteins; and (3) a contextual encoder for compound encoding and refinement.

The compound decoder was pre-trained on 10 million SMILES randomly sampled from PubChem. The compound decoder adopts the autoregressive pre-training objective used in GPT, aiming to predict the next SMILES token based on preceding tokens (Fig. 1a). This training strategy allows for the sequential generation of compounds in both unconditional and conditional manners, depending on whether target information is provided or not. With this pre-training strategy, TamGen is able to learn general and diverse knowledge about a multitude of compounds from chemical databases (e.g., PubChem), without requiring any additional information such as binding proteins. This strategy enhances the generation capability of the compound

decoder and improves the chemical properties of the generated compounds.

The protein encoder was developed to comprehend target protein information and to facilitate the generation of drug-like compounds in a target-aware manner (Fig. 1b left). The Transformer architecture adopted by the protein encoder features a self-attention mechanism, which gathers and processes information from input sequences. Here, we designed a variant of self-attention to capture both the sequential and geometric data of target proteins (Fig. S1, see Methods for details). The protein encoder's outputs are then directed to the compound decoder via a cross-attention module (Fig. 1c), activated only when target proteins are provided. Therefore, we are able to generate compounds from the 3D conformation of target proteins via the protein encoder-compound decoder framework.

A Variational Autoencoder (VAE)-based contextual encoder was employed to encode compounds and assist the generation process. VAEs are commonly used to create new data by learning the input data's probability distribution and sampling from it[38]. In TamGen, the VAE-based contextual encoder determines the mean ($\mu$) and standard deviation ($\sigma$) for any given compound **y** and protein sequence **x** pair (Fig. 1b right). Later, a vector $z$ is sampled from the distribution determined by $\mu$ and $\sigma$ and added to the output of protein encoder, before directed to the compound decoder (Fig. 1b right). In the training stage, the model's objective is to recover the input compound **y**, whereas during application, the contextual encoder facilitates compound refinement once a seeding molecule is provided. The incorporation of this encoder enhances control over compound generation, enabling TamGen to be seamlessly integrated into multi-round drug optimization pipelines with human feedback. This interactive and iterative drug design capability holds the potential to increase the success rate of designed compounds and accelerate the drug discovery process.

### TamGen is effective and efficient for generative drug design
To benchmark the overall performance of TamGen, we compared our methods against five approaches proposed recently: liGAN[39], 3D-AR[40] (there is no abbreviation for the proposed method, so we refer to it as 3D-AR), Pocket2Mol[41], ResGen[42] and TargetDiff[19]. These approaches focus on direct generation of compounds in the 3D space to match protein binding pockets with diverse deep learning techniques. Following previous practices, we evaluated these methods and TamGen on CrossDocked2020 dataset[43], a well-established benchmark dataset curated from PDBbind. CrossDocked2020 is composed of a train set with about 100,000 drug-target pairs and a test set with 100 protein binding pockets. For fair comparison with previous work, we used the same training and test data as those used in[19,41] to fine-tune TamGen.

We generated 100 compounds for each target protein in CrossDocked2020 test set with each method respectively. Then, we evaluated the designed compounds using a comprehensive set of metrics: binding affinity to target proteins, estimated by docking scores from Autodock-Vina[44]; drug-likeness, assessed using both the Quantitative Estimate of Drug-likeness (QED)[45] and Lipinski's Rule of Five[46] based on calculated molecular physicochemical properties; synthetic accessibility scores (SAS), estimated by RDKit as a proxy for the ease of synthesis of a compound[47]; and LogP, an indicative of molecular lipophilicity, with an optimal range of 0–5 for oral administration[48]. In addition, we quantified the ability to generate diverse compounds of each method with molecular diversity. Molecular diversity is derived from the Tanimoto similarity between Morgan fingerprints of compounds. This set of metrics provides a broad and complementary assessment of compound properties, indicating the overall efficacy of a drug design method.

While each method demonstrates strengths across certain metrics, TamGen is consistently top ranked. For example, TamGen achieves either the first or the second place in 5 out of 6 metrics and

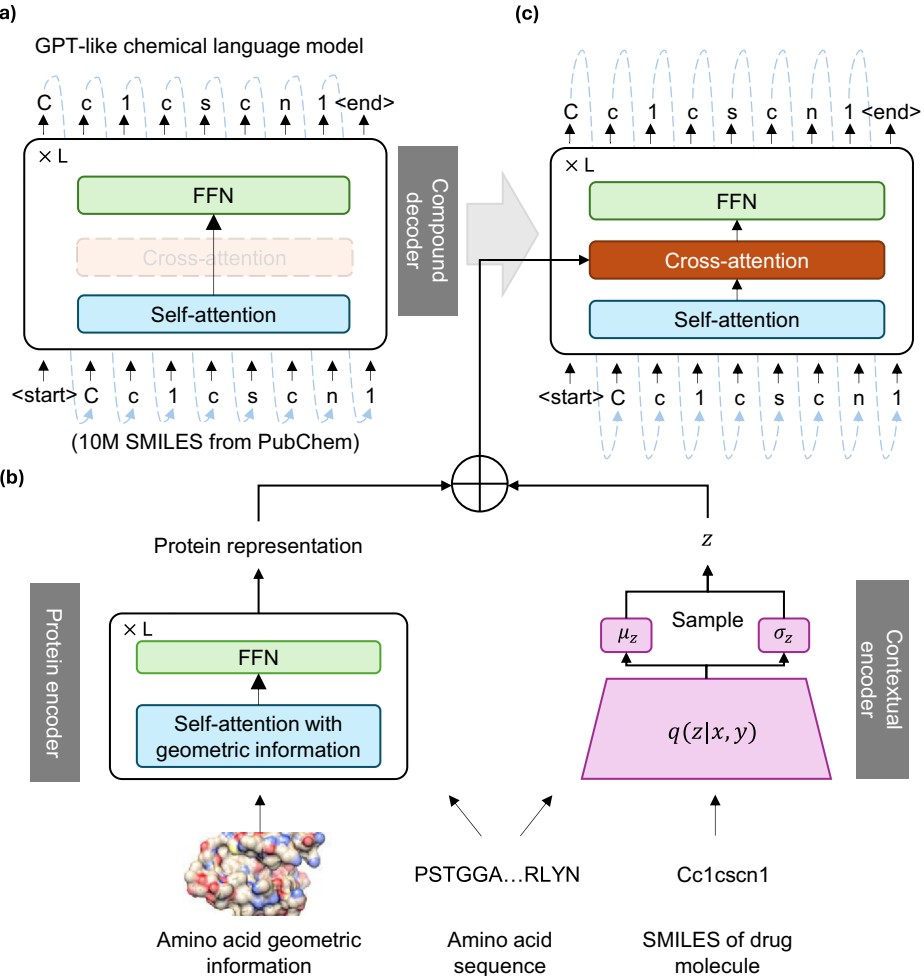

**Fig. 1 | The architecture of TamGen. a** The pre-training phase of the compound decoder, a GPT-like chemical language model. The model adopts standard GPT architecture, which autoregressively generates the SMILES tokens, 1D molecular string representations of compounds, from the input. 10 million compounds randomly selected from PubChem were used for pre-training. **b**, **c** The overall framework of TamGen during the fine-tuning and inference stages. **b** A Transformer-based protein encoder and a VAE-based contextual encoder to facilitate target-aware drug generation and seeding molecule-based compound refinement. See Methods and Fig. S1 for details. **c** The outputs from the protein encoder and the contextual encoder are integrated and forwarded to the compound decoder via a cross-attention module. 1D molecular string representations of the compounds in SMILES are then generated by our model.

exhibits the best overall performance (Fig. 2a, Fig. S2 and Table S1). This finding shows that TamGen is capable of simultaneously optimizing multiple aspects of compounds during the generation process.

Among the metrics, synthetic accessibility is an important factor affecting the practicality of a drug candidate, especially for novel compounds. It is worth pointing out that TamGen performs the best in terms of SAS for compounds with high binding affinity (reflected on docking scores, Fig. 2b), which are likely to possess superior bioactivity against target proteins. To discern why TamGen generates compounds with both high binding affinity and favorable SAS, we examined the top-scoring compounds generated by TamGen and other methods. Our analysis reveals that TamGen tends to produce compounds with fewer fused rings (Fig. 2c and Fig. S3). Notably, the number of fused rings in compounds generated by TamGen aligns closely with FDA-approved drugs, averaged to 1.78 (Fig. 2c and Fig. S3). Conversely, while methods involving direct 3D generation can sometimes create compounds with superior poses within binding pockets, these compounds often feature multiple fused rings (Fig. 2c–d). Prior research indicates that a higher number of fused rings may lead to lower SAS[49–51], potentially accounting for the subpar SAS scores of other methods. Moreover, a high count of fused rings is linked with increased cellular toxicity and decreased developability[51,52]. In line with this understanding, compounds generated by TamGen display a

higher similarity score to FDA-approved drugs (Fig. S4). We hypothesize that pre-training on natural compounds and employing a sequence-based generation strategy enhance the overall plausibility of compounds produced by TamGen.

TamGen also achieves the best efficiency compared to alternate methods (Fig. S5). We benchmarked the wall time to generate 100 compounds for each target of all methods using one A6000 GPU. Other methods required tens of minutes or hours to complete this task, while TamGen was able to accomplish the task in an average time of just 9 seconds. This makes TamGen 85, 154, 213 and 394 times faster than ResGen, TargetDiff, Pocket2Mol and 3D-AR, respectively.

Collectively, our results suggest that TamGen is both effective and efficient in generating novel compounds. This positions TamGen as a valuable asset for quickly identifying hit compounds for downstream development.

## TamGen designs novel inhibitors targeting Tuberculosis ClpP protease

We next employed TamGen to design small-molecule inhibitors against ClpP. As mentioned, ClpP plays essential roles in maintaining bacterial homeostasis, rendering it a promising antibiotic target.

Apart from the previously identified Bortezomib, a peptidomimetic compound that targets the human 26S proteasome and exhibits

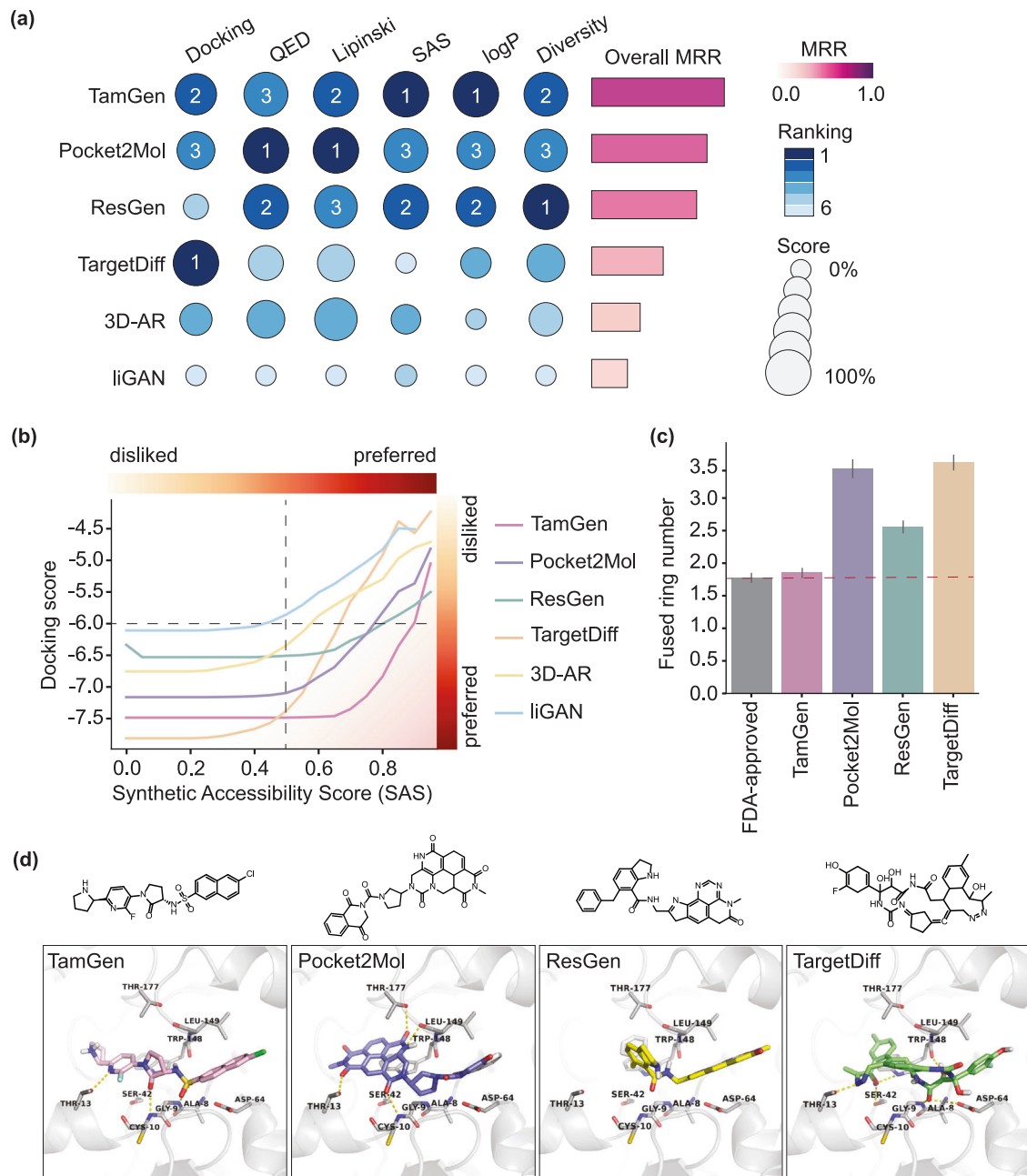

**Fig. 2 | TamGen achieves the state-of-the-art performance on compound generation. a** Overview of generative drug design methods ranked by overall scores for the CrossDocked2020 task. Left: Metrics include docking score (lower scores indicate better binding affinity), quantitative estimation of drug-likeness (QED), Lipinski's Rule of Five, Synthetic accessibility scores (SAS), LogP, and molecular diversity (Div). Sizes of dots: scores (mean). Darkness of dots: rankings. Scores were normalized to 0%–100% for each metric. Absolute values were used for docking score normalization. The original data used for plotting can be found in Table S1 and Fig. S2. Right: The overall score for each method was calculated with mean reciprocal rank (see Methods for details). **b** Average docking scores against SAS for TamGen and alternate methods. TamGen achieves more favorable docking scores

for compounds with higher SAS and lower docking scores (bottom-right corner). **c** Barplot of the number of fused rings (see Methods for details) in FDA-approved drugs and top-ranked compounds generated by selected methods. For each method, a statistics of 1,000 compounds (100 targets ×10 compounds with the highest docking scores against each corresponding target) were plotted. The dashed line represents the average number of fused rings in FDA-approved drugs. Data is presented as mean values ± 95% confidence interval. **d** Example compounds generated by selected methods, and their binding poses to one target protein (shown as ribbons, with key residues shown as sticks). Source data are provided as a Source Data file and can also be accessed in the Zenodo repository of TamGen[71].

inhibitory activity against bacterial ClpP[53,54], there are currently no documented advanced antibiotic ClpP inhibitors. Therefore, we leverage TamGen to generate compounds targeting ClpP in *Mycobacterium tuberculosis* (Mtb), a pathogenic bacteria in urgent need for novel drug candidates.

We adopted a Design-Refine-Test pipeline driven by TamGen to identify potential ClpP inhibitors (Fig. 3). During the Design stage

(Fig. 3a), utilizing the binding pocket of ClpP derived from protein structures (PDB ID 5DZK), TamGen generated 2612 unique compounds.

These compounds were then screened using molecular docking and Ligandformer, an AI model for phenotypic activity prediction[55] (see Methods for details). At this stage, we eliminated the compounds with worse docking scores compared to Bortezomib and inactive

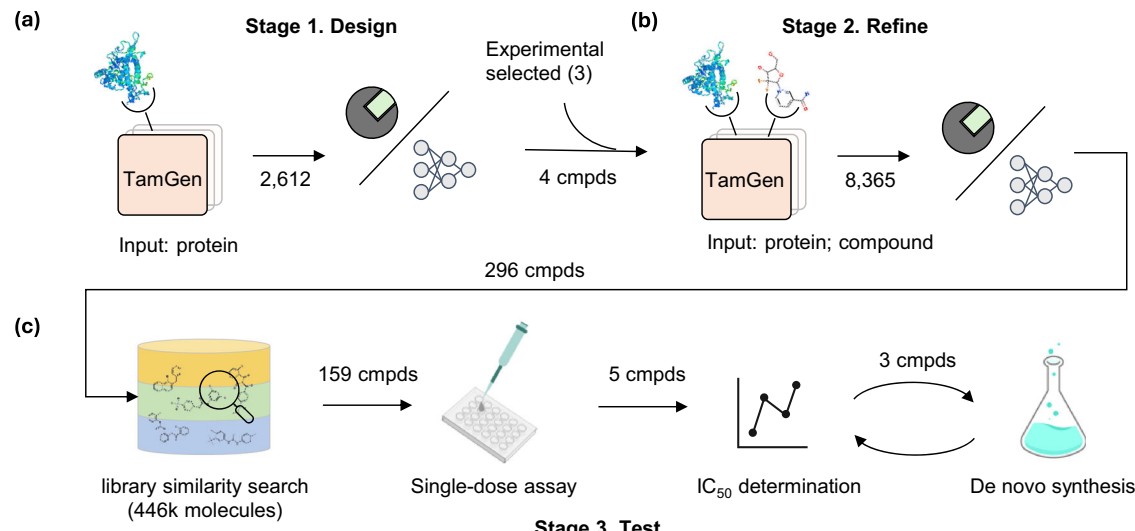

**Fig. 3 | Illustration of the Design-Refine-Test pipeline for Tuberculosis drug generation. a** The Design stage. **b** The Refine stage. **c** The Test stage.

compounds predicted by Ligandformer. Peptidomimetic compounds were also excluded due to their suboptimal ADME properties (which is a known drawback of Bortezomib[56]). Finally, we identified 4 seeding compounds (Fig. S6) for the following Refine stage.

In the Refine stage, TamGen was applied to generate compounds conditioned on both the target protein and seeding compounds (Fig. 3b). Here, in addition to the 4 representative compounds generated by TamGen, we included 3 compounds with weak inhibitory activities (IC$_{50}$ in 100–200 μM against Mtb ClpP.) identified from previous experiments (Fig. S6, see Supplementary notes for details). Conditioned on the ClpP and these 7 seeding compounds, we generated 8,635 unique compounds using TamGen, and screened the compounds following the same procedure as in the Design stage. Finally, 296 of these generated compounds were selected for the Test (biological assay) stage.

We proceeded to compare the generated compounds with molecules from existing chemical libraries. Using UMAP visualization (Fig. 4a, See Methods for details), we observe that compounds generated by TamGen are distinguishable from those in compound libraries. This indicates that TamGen is capable of exploring untapped chemical spaces when generating potential compounds conditioned on ClpP. Moreover, the compounds generated in the Refine stage showed superior docking scores and more dispersed patterns (an indicative of molecular diversity) compared to those from the Design stage (Fig. S7). This improvement shows that a Design-Refine generation approach can effectively enhance the desired properties of the candidate pool.

### TamGen-driven drug design yields effective inhibitors against Tuberculosis ClpP protease

To expedite the validation process and enhance the efficiency during the Test stage, we first sought commercially available compounds structurally akin to those generated by TamGen (Fig. 3c). From a 446k commercial compound library, we successfully pinpointed 159 analogs with Maximum Common Substructure (MCS) similarity scores exceeding 0.55 in comparison to any of the 296 selected TamGen compounds. Five of these analog compounds displayed significant inhibitory effects in the ClpP1P2 peptidase activity assay, with Bortezomib serving as a positive control (Fig. S8). Subsequent dose-response experiments revealed IC$_{50}$ values below 20 μM for all five compounds, with Analog-005 standing out with an IC$_{50}$ of 1.9 μM(Fig. 4b). Notably, none of these compounds have been previously documented as ClpP inhibitors (Table S2).

To explore the structure-activity relationship (SAR) and expand the pool of hit compounds, we synthesized three novel compounds absent from the commercial library. Considering that Analog-003 exhibited the strongest inhibitory effect in the peptidase activity assay (48% of Bortezomib, Fig. S8), we first synthesized its corresponding source compound generated by TamGen, referred to as Syn-A003-01 (Fig. 4a). Both compounds, along with Analog-001 and Analog-002, share a diphenylurea core (Series I in Fig. 4a), representing a novel scaffold for ClpP inhibitors. Interestingly, dose-response assay showed that replacing trifluoromethyl with chlorine greatly improved inhibitory activity of the compound (Fig. 4b). We reason that the replacement may have altered the charge distribution of the adjacent urea group in the compound, thereby influencing its hydrogen bonding effects. In addition, substituting sulfonamide group with fluorine also moderately improved the activity. Secondly, we synthesized two derivatives of Analog-005, the compound with the most favorable IC$_{50}$ (Fig. 4a, Series II). Similar inhibition efficiency was observed in these two derivatives and Analog-005 (Fig. 4b). This result suggests a marginal contribution to the overall activity from the modified groups and provides the starting point for further modifications. All compounds generated or inspired by TamGen displayed noteworthy IC$_{50}$ values. The high confirmation rate of TamGen-driven drug design also highlights an alternative application of generative models, specifically employing the newly generated molecules as anchors for a more effective and efficient library search. This approach allows us to alleviate the cost in screening process and surmount the challenges posed by the validation and application of novel molecule synthesis in generative methods.

### Structural insights on the mechanisms of compound binding

To investigate the inhibitor binding mechanism, we analyzed the docking poses of two representative compounds, Syn-A003-01 (from Series I) and Analog-005 (from Series II). These two compounds were docked to ClpP structure (PDB ID: 5DZK, see Methods for details) (Fig. 5). For comparison, the binding pose of Bortezomib was also aligned into the same crystal structure of ClpP. Similar to Bortezomib, both Analog-005 and Syn-A003-01 maintain multiple hydrogen bonding interactions with ClpP1 (a subunit of ClpP). Meanwhile, the docked pose of Analog-005 suggests that the carbonyl carbon possibly forms a covalent bond with the catalytic residue Ser98, as indicated by both the chemical mechanism and docked complex structural model. This is in accordance with the binding pose of Bortezomib, providing plausible explanation of Analog-005's strong inhibitory activity.

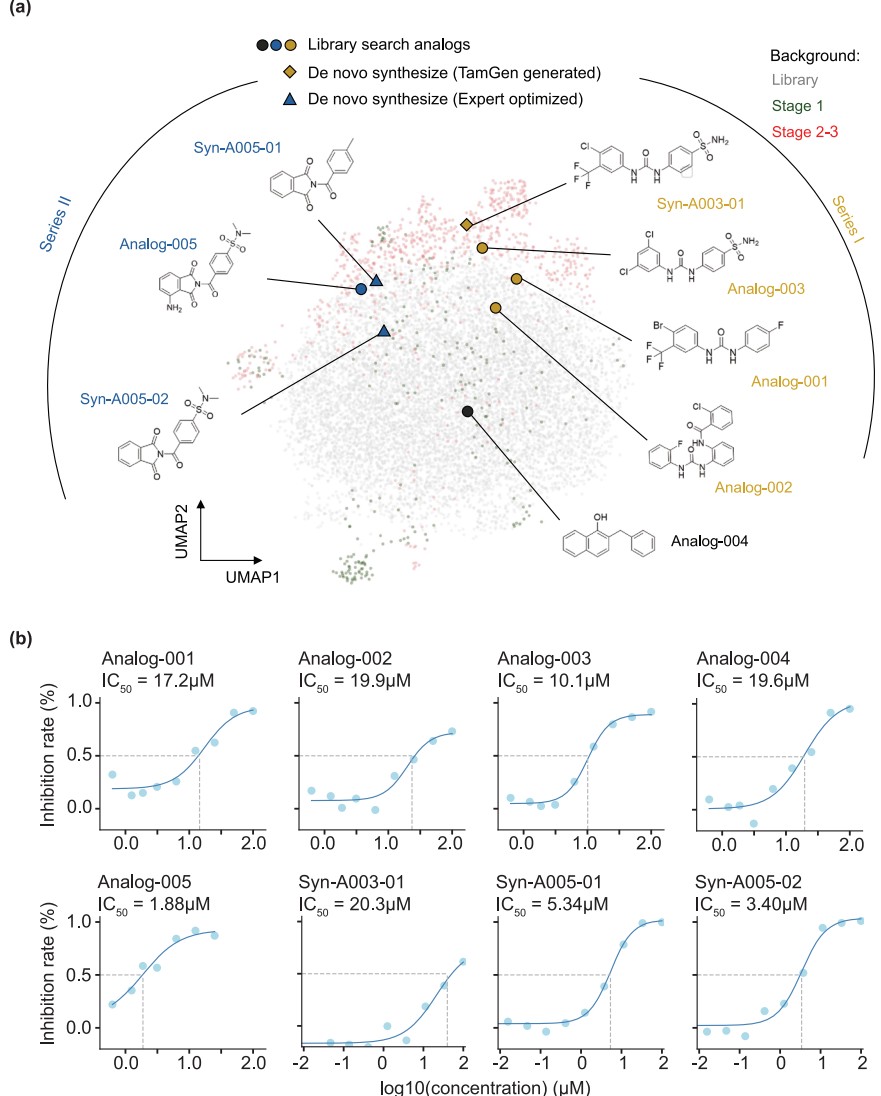

**Fig. 4 | Visualization and experimental validation on designed compounds.**
**a** UMAP visualization of library compounds and key compounds identified from the Design-Refine-Test pipeline with TamGen. Gray (background): compounds sampled from library. Green (background): Sampled compounds generated at Stage 1. Red (background): Sampled compounds generated at Stage 2. Circle, triangle, and diamond markers: compounds subjected to IC$_{50}$ determinations, with different shapes indicating different compound sources. These compounds were further stratified into 3 clusters (series I: yellow, series II: blue, and others: black) based on their molecular scaffold groups. **b** Dose-response assays for eight compounds with DMSO as a control. See methods for details of curve fitting and IC$_{50}$ determination. Source data are provided as a Source Data file.

Interestingly, the complex structures also reveal that the sulfonamide groups of Analog-005 and Syn-A003-01 extend towards a deep pocket formed by residues Glu101, Phe102, Met150 and Asn154, a feature not observed for Bortezomib. The sulfonamide group may contribute to the binding to ClpP.

Altogether, through the Design-Refine-Test process powered by TamGen, we identified compounds that interact with ClpP protein in distinct modes from that of Bortezomib, thereby unveiling novel mechanisms for future ClpP inhibitor discovery. These compounds possess benzenesulfonamide and diphenylurea groups as scaffolds, which are completely different from the peptidomimetic Bortezomib, providing a possible solution to improve bioavailability and molecular stability of ClpP inhibitors. To sum up, the novelty and strong inhibitory efficacy of these compounds show potential for further development. The success of generating ClpP inhibitory compounds underscores the immense promise of TamGen in designing novel drug candidates and addressing drug-resistant Tuberculosis, implying its broad applications in drug design to treat other diseases.

## Investigate TamGen's performance through ablation experiments

To elucidate the factors contributing to TamGen's superior performance in both benchmark tests and real-world applications, we conducted a series of ablation experiments.

First, we examined the necessity of pre-training for generating plausible chemical compounds. Using the same training/testing split of the CrossDocked2020 dataset as in Fig. 2, we observed that TamGen models without pre-training resulted in significantly poorer docking scores (Fig. S9a) and generated compounds with simpler structures (Fig. S9b), which are suboptimal for drug candidate development.

Next, we evaluated TamGen's ability to capture and utilize pocket-ligand pair information during fine-tuning by randomly shuffling the pocket-ligand pairs in the CrossDocked2020 training dataset and re-training TamGen (referred to as TamGen-r, see Methods for details). Consistent with our expectations, the correctly paired data version of TamGen achieved notably better docking scores compared to

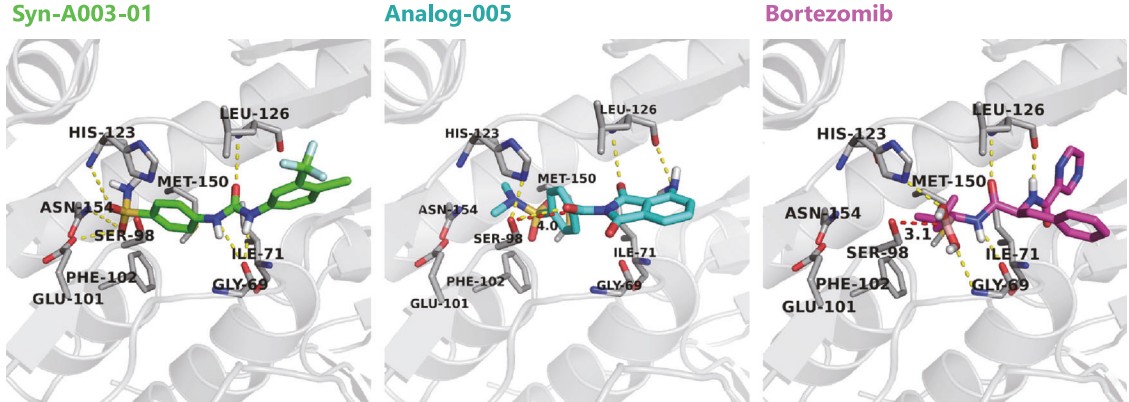

**Fig. 5 | Proposed binding modes of Syn-A003-01, Analog-005, and Bortezomib against ClpP.** ClpP complex 5DZK is presented in grey cartoon. Syn-A003-01, Analog-005, and the reference compound Bortezomib are shown in green, cyan, and magenta sticks, respectively. The yellow dashed lines indicate hydrogen bonds. The red dashed lines with numbers denote distances between atoms. Source data of the docking poses are in the Zenodo repository of TamGen[71].

TamGen-r (Fig. S10), underscoring the importance of accurate pocket-ligand interactions.

We then investigated the impact of the self-attention variant in TamGen on its performance (see Methods for details). The results (Table S3) indicate a significant decline in docking scores upon the exclusion of distance-aware attention and coordinate data augmentation. This highlights the critical role of these techniques in capturing essential protein information. Given the lack of pre-training for the protein component and the scarcity of protein data, incorporating data augmentation and inductive biases, such as the assumption that proximate residues are more likely to interact, proves beneficial.

Finally, to demonstrate TamGen's capability to generate active compounds against targets, we further synthesized 8 compounds directly generated by TamGen against ClpP bypassing the library search step (see Methods for details). Remarkably, 6 out of the 8 compounds from 6 distinct structural clusters demonstrated an $IC_{50}$ below 40 μM (Fig. S11), showcasing TamGen's effectiveness in producing a variety of promising hits for the specified targets.

## Discussion

Designing compounds with high binding affinities to pathogenic protein targets can significantly accelerate the drug discovery process. Generative AI methods that generate compounds based on target information contribute not only by expediting this process but also by enabling the exploration of a larger chemical space beyond existing compound libraries, rendering it a highly desirable goal. However, despite the considerable efforts on generative AI methods, few attempts have demonstrated success in real-world applications. Here, we present our method, TamGen, which not only achieved state-of-the-art performance in benchmark testing but also discovered several compounds with high inhibitory activities against the ClpP protease of *Mycobacterium tuberculosis*, the causative pathogen of tuberculosis.

The success of TamGen can be attributed to three major factors. First, the chemical knowledge embedded in the pre-trained compound decoder model enables the generation of high-quality compounds that adhere to chemistry rules and possess properties conducive to drug development. Second, an effective binding pocket representation that incorporates both sequential and geometric information supports robust chemical compound generation. As demonstrated in previous sections, these design elements contribute to generating compounds with high diversity and drug-likeness properties, thereby increasing the chances of identifying compounds that can be synthesized and developed into drugs. Lastly, a Variational Autoencoder (VAE)-based contextual decoder allows for the refinement of hit compounds using candidate molecules reported in the literature or identified in previous rounds, generating improved compounds for given targets. Leveraging this feature, we adopted a Design-Refine-Test pipeline for iterative compound generation and enhancement. This pipeline was instrumental in the successful design of strong inhibitor compounds against the Mtb ClpP, where we first generated compounds de novo with TamGen and then refined the selected compounds following preliminary filtering. The refinement stage can be repeated multiple times by including inhibitors discovered in previous steps, allowing TamGen to further optimize the compounds and increase the likelihood of generating better inhibitors.

Target-based de novo drug generation methods, such as TamGen, have the unique advantage of generating novel compounds that extend beyond existing chemical libraries. However, this novelty also presents certain challenges. New compounds often lack comprehensive data on their in vivo properties, such as toxicity, metabolism, and pharmacokinetics. Moreover, even if these compounds are highly synthesizable, the time required for synthesis can be a limiting factor when compared to the immediate availability of compounds from existing libraries. To mitigate these issues and harness the strengths of both generative AI and traditional screening methods, we implemented a library search step in the pipeline. Instead of immediately synthesizing the generated compounds, we first searched for analogs in commercially available libraries using a loose similarity threshold and tested their biological activity. Subsequently, we synthesized the generated compounds and conducted structure-activity relationship (SAR) analyses based on structurally related compounds. This approach thus can balance the novelty and potential of AI-generated compounds with the practicality and speed of analog testing, thereby streamlining the drug discovery pipeline.

However, TamGen is not without its limitations. For instance, it is not sensitive enough to distinguish targets with minor differences, such as point mutations or protein isoforms (data not shown), which could be critical for drug design targeting cancer-related proteins and other conditions. Additionally, as a structure-based drug design approach, the application of TamGen requires the structure of the target protein and potential binding pocket information. With the advent of tools like AlphaFold series[57,58] for protein structure predictions and improved pocket identification methods, the barriers to TamGen's application are expected to reduce over time[59].

Recently, there has been an increasing focus on target-aware molecule generation in three-dimensional (3D) space. These approaches can be broadly categorized into diffusion-based methods[19–22] and autoregressive methods[12–14,18,40,41]. While we have demonstrated that current 3D generation methods, which may be limited by the availability of high-quality protein-ligand complex structures, tend to

generate molecules with less ideal drug-likeness features, it is important to acknowledge that 3D interactions between compounds and proteins are more fundamental for structure-based drug design. Therefore, 1D generation using SMILES and amino acid sequences, such as TamGen, may not fully exploit the richer interactions available in geometric space. In the future, we aim to integrate insights from 3D generation methods, particularly autoregressive approaches, with the current designs of TamGen to improve compound generation and refinement, potentially yielding compounds with better docking scores and more reasonable structures.

In addition, alternative 1D generation approaches, such as AlphaDrug[15], has achieved impressive docking scores by combining transformers with a Monte Carlo Tree Search (MCTS) algorithm guided by docking scores, thereby optimizing the properties of generated molecules (Table S4). MCTS presents a robust strategy for guiding molecule generation in the absence of explicit supervised signals. Looking forward, we aim to enhance TamGen by either using the MCTS process or employing reinforcement learning techniques to guide molecule generation for better docking scores and other drug properties, such as compound stability, synthesizability, and ADME/T properties. A recent work, PrefixMol[60], also aims to generate SMILES based on both the pocket information and the compound properties, such as QED and SA scores. Overall, the property guided generation points to an important direction for future development.

Lastly, because TamGen primarily focuses on hit identification and expansion, we have not extensively tested the cellular activities and toxicities of the proposed compounds. To advance further down the drug discovery pipeline, additional evaluation and optimization of these candidates will be necessary. Despite areas for improvement, TamGen's ability to generate novel compounds based solely on the structure and pocket information of target proteins makes it a powerful and user-friendly tool for structure-based drug design. This flexibility suggests that TamGen can be effectively employed in the discovery of therapeutics for a wide range of targets, and we are also exploring TamGen's applicability across various biological scenarios. As we gather more experimental validation data for the compounds generated by AI models, this additional information will enhance future generative AI models, including TamGen. We expect that this iterative process will significantly amplify their potential in facilitating global drug discovery efforts.

## Methods

### Details of TamGen

We describe the details about how to process the 3D structure input, the architectures of the protein encoder, the chemical language model, the contextual encoder and the training objective functions.

*Preliminaries*: Let $\mathbf{a} = (a_1, a_2, \cdots, a_N)$ and $\mathbf{r} = (r_1, r_2, \cdots, r_N)$ denote the amino acids and their 3D coordinates of a binding pocket respectively, where $N$ is the sequence length and $r_i \in \mathbb{R}^3$ is the centroid of amino acid $i$ ($i$ is an index to label the amino acids around the binding site). $a_i$ is a one-hot vector like $(\cdots, 0, 0, 1, 0, \cdots)$, where the vector length is 20 (the number of possible amino acid types) and the only 1 locates at the position corresponding to the amino acid type. A binding pocket is denoted as $\mathbf{x} = (\mathbf{a}, \mathbf{r})$ and $[N] = \{1, 2, \cdots, N\}$. Let $\mathbf{y} = (y_1, y_2, \cdots, y_M)$ denote the SMILES string of the corresponding ligand/compound with a length $M$. Our goal is to learn a mapping from $\mathbf{x} = (\mathbf{a}, \mathbf{r})$ to $\mathbf{y}$.

*Processing 3D input*: The amino acid $a_i \forall i \in [N]$ is mapped to $d$-dimensional vectors via an embedding layer $E_a$. Following our previous exploration on modeling the 3D coordinates[61], the coordinate $r_i (i \in [N])$ is mapped to a $d$-dimensional vector via a linear mapping. Considering we can rotate and translate a binding pocket while its spatial semantic information should be preserved, we apply data augmentation to the coordinates. That is, in the input layer, for any

$i \in [N]$,

$$h_i^{(0)} = E_a a_i + E_r \rho\left(r_i - \frac{1}{N}\sum_{j=1}^{N} r_j\right), \tag{1}$$

where (i) $E_a$ and $E_r$ are learnable matrices, and they are optimized during model training; (ii) $\rho$ denotes a random roto-translation operation, and before using $\rho$, we center the coordinates to the origin. Thus we process the discrete input $\mathbf{x}$ into $N$ continuous hidden representations $h_i^{(0)}$.

*Protein encoder*: The encoder stacks $L$ identical blocks. The output of the $l$-th block, i.e., $h_i^{(l)}$, is fed into the $(l + 1)$-th layer for further processing and obtain $h_i^{(l+1)}$ for any $i \in [N]$ and $l \in \{0\} \cup [L - 1]$. Each block consists of an attention layer and an FFN layer, which is a two-layer feed-forward network as that in the original Transformer[25]. To model the spatial distances of amino acids, we propose a new type of distance-aware attention. Mathematically,

$$\tilde{h}_i^{(l+1)} = \sum_{j=1}^{N} \alpha_j (W_v h_j^{(l)}),$$
$$\alpha_j = \frac{\exp \hat{\alpha}_j}{\sum_{k=1}^{N} \exp \hat{\alpha}_k}, \tag{2}$$
$$\hat{\alpha}_j = \exp\left(-\frac{\|r_i - r_j\|^2}{\tau}\right)(h_i^{(l)\top} W h_j^{(l)}),$$

where $W$ and $W_v$ are parameters to be optimized, and $\tau$ is the temperature hyperparameter to control. After that, $\tilde{h}_i^{(l+1)}$ is processed by an FFN layer and obtain

$$h_i^{(l+1)} = \mathrm{FFN}\left(\tilde{h}_i^{(l+1)}\right). \tag{3}$$

The output from the last block, i.e., $h_i^{(L)} \forall i \in [N]$, is the eventual representations of $\mathbf{x}$ from the encoder.

*The contextual encoder*: To facilitate diverse generation, we follow the VAE framework and use a random variable $z$ to control the diverse generation for the same input. Given a protein binding pocket $\mathbf{x}$, a compound $\mathbf{y}$ is sampled according to the distribution $p(\mathbf{y}|\mathbf{x}, z; \Theta)$. The contextual encoder (i.e., the VAE encoder) models the posterior distribution of $z$ given a binding pocket $\mathbf{x}$ and the corresponding ligand $\mathbf{y}$. The input of VAE encoder is defined as follows:

$$h_i^{(0)} = \begin{cases} E_a a_i + E_r \rho\left(r_i - \frac{1}{N}\sum_{j=1}^{N} r_j\right), & i \leq N \\ E_y y_{i-N}, & i > N, \end{cases} \tag{4}$$

where $E_y$ is the embedding of the SMILES. The VAE encoder follows the architecture of standard Transformer encoder[25], which uses the vanilla self-attention layer rather than the distance-aware version due to the non-availability of the 3D ligand information. The output from the last block, i.e., $h_i^{(L)} \forall i \in [N]$, is mapped to the mean $\mu_i$ and covariance matrix $\Sigma_i$ of position $i$ via linear mapping, which can be used for constructing $q(z|\mathbf{x}, \mathbf{y})$, by assuming $q(z|\mathbf{x}, \mathbf{y})$ is Gaussian. The ligand representations, i.e., $h_j^{(L)} j > N$, are not used to construct $q(z|\mathbf{x}, \mathbf{y})$.

*Chemical language model*: The chemical language model is exactly the same as that in ref. 25, which consists of the self-attention layer and the FFN layer. We pre-train the decoder on $10M$ compounds randomly selected from PubChem (denoted as $\mathcal{D}_0$) using the following objective function:

$$\min - \sum_{y \in \mathcal{D}_0} \frac{1}{M_y} \sum_{i=1}^{M_y} \log P(y_i | y_{i-1}, y_{i-2}, \cdots, y_1), \tag{5}$$

where $M_y$ is the length of $y$. The chemical language model is pre-trained on eight V100 GPUs for 200k steps.

After pre-training the chemical language model, the cross-attention module is introduced to the compound decoder as shown in Fig. 1c (top panel). It takes all $h_i^{(L)}$ as inputs. Under the VAE variant, during training and compound refinement, the inputs are $h_i^{(L)} + z_i'$, where $z_i'$ is sampled from the distribution $q(z|\mathbf{x}, \mathbf{y})$ introduced above. During inference, the inputs are $h_i^{(L)} + z_i$ where $z_i$ is randomly sampled from $\mathcal{N}(0, I)$.

*Training*: The training objective is to minimize the following function:

$$\min_{\Theta, q} \frac{1}{|\mathcal{D}|} \sum_{(\mathbf{x}, \mathbf{y}) \in \mathcal{D}} -\log P(\mathbf{y}|\mathbf{x}, z; \Theta) + \beta \mathcal{D}_{kl}(q(z|\mathbf{x}, \mathbf{y}) \| p(z)). \quad (6)$$

In Eq. (6), $\mathcal{D}$ is the training corpus, a collection of (pocket, SMILES) pairs; $z$ in $\log P(\cdots)$ is sampled from $q(z|\mathbf{x}, \mathbf{y})$; $\beta$ is a hyperparameter; $p(z)$ denotes the standard Gaussian distribution; $\mathcal{D}_{kl}$ denotes the KL divergence; $\Theta$ denotes the parameter. The first term in Eq. (6) enables the model to learn how to generate a reasonable ligand based on the input pocket. The second term, the KL divergence constraint, ensures that the learned latent space resembles the prior distribution $p(z)$, which promotes smooth interpolation and generalization. It regularizes the encoder by pulling the approximate posterior distribution (the encoder's output) closer to the prior distribution (typically a simple Gaussian). This helps in preventing overfitting and ensures meaningful latent representations.

*Implementation details* For the results in Fig. 2, for fair comparison with the previous methods like Pocket2Mol[41], Targetdiff[19], we use the same data as them. The data is filtered from CrossDocked[43] and there are about 100*k* target-ligand pairs. For inference, the $z$ is sampled from multivariant standard Gaussian distribution. Both the pocket encoder and VAE encoder have 4 layers with hidden dimension 256. The decoder has 12 layers with hidden dimension 768. We use Adam optimizer[62] with initial learning $3 \times 10^{-5}$. In the context of generating the compound database for Tuberculosis (TB), the current methodology incorporates an augmented dataset that includes the Cross-Docked database and the Protein Data Bank (PDB), cumulatively accounting for approximately 300,000 protein-ligand pairs. To elaborate, this process involved the extraction of pocket-ligand pairs from about 72,000 PDB files. A pocket is defined on the basis of spatial proximity criteria: if any atom of an amino acid is less than 10Å away from any atom of the ligand, the corresponding amino acid is taken as part of the pocket.

### The phenotype screening predictor Ligandformer

We utilized an adapted version of the Graph Neural Network (GNN) model as proposed in ref. 55 to predict potential phenotypic activity. Compared with traditional GNNs, our model is designed such that the output from one layer is propagated to all subsequent layers for enhanced processing. We implemented a 5-layer architecture. Our phenotypic predictor was trained using a dataset of 18,886 samples, which are gathered from a variety of sources including ChEMBL, published datasets, and academic literature as compiled by ref. 63. At the inference stage, we interpreted an output value exceeding 0.69 (a threshold determined based on validation performance) as indicative of a positive sample.

### Baselines and evaluations

**Baselines.** We mainly compare our method with the following baselines:

1. 3D-AR[40], a representative deep learning baseline that uses a graph neural network to encode the 3D pocket information and direct generates the 3D conformation of candidate drugs. The atom type and coordinates are generated sequentially. 3D-AR does not explicitly generate the position of the next, by use MCMC for generation.
2. Pocket2Mol[41] is an improved version of 3D-AR, which has specific modules to predict atom type, coordinate positions and bond type.
3. ResGen[42] is also an autoregressive method of generating compounds in 3D space directly. Compared with Pocket2Mol, ResGen uses residue-level encoding while Pocket2Mol uses atomic-level encoding.
4. TargetDiff[19] utilizes diffusion models to generate compounds. Compared with the previous method, all atom types and coordinates are generated simultaneously, and iteratively refined until obtaining a stable conformation.

**TamGen without pre-training.** To assess the impact of pre-training, we introduce a TamGen version without pre-training, in which the compound generator is initialized randomly. We observed overfitting when a 12-layer chemical language model was used in the non pre-trained version. Upon evaluating layers 4, 6, 8, and 12 based on their validation performance, we discovered that a model with 4 layers yielded the most optimal results.

**Mean reciprocal rank (MRR).** Mean Reciprocal Rank (MRR) calculation[64] is a widely used method to evaluate a method across different metrics. To elaborate, denote the rank of a method on metric $i$ as $r_i$. The MRR for a particular method is hence defined as $\frac{1}{N}\sum_{i=1}^{N}\frac{1}{r_i}$, where $N$ represents the total number of evaluation metrics being considered.

**Fused rings.** In this work, *fused rings* denote a structural element in compounds where two or more ring structures share at least one common bond. The size of the largest group of these "fused" rings within a molecule is denoted as the number of fused rings. In Fig. 2d, from left to right, the number of fused rings of the four compounds are 2, 5, 4, and 4, respectively.

### Experimental details

**Peptidase activity assay.** ClpP1P2 complex in Mtb can catalyze the hydrolysis of small peptides. Following previous protocols, we measure the in vitro inhibition of ClpP peptidase activity by monitoring the cleavage of fluorogenic peptide Ac-Pro-Lys-Met-AMC[65–67].

0.4 μL of candidate inhibitors, Bortezomib, or DMSO control are added into a black flat bottom 384-well plate by Echo®20 Liquid Handler and mixed with 20 μL enzyme buffer (The final ClpP1P2 dimer concentration is 50nM; reaction buffers: PIPES 30mM (pH 7.5), NaCl: 200mM and 0.005% Tween20). The solution is pre-incubated at room temperature for 2 hours. Then, 20 μL substrate buffer with Ac-Pro-Lys-Met-AMC is added (final concentration of Ac-Pro-Lys-Met-AMC is 10 μM; reaction buffer is the same with the above). Fluorescence (Ex/Em: 380/ 440 nm) is recorded for 120 min at 37 °C.

**Single-dose response measurement.** Inhibition rates of compounds were determined by Relative Fluorescence Units (RFU) compared with Bortezomib control[68,69] and DMSO control, which is defined as follows:

$$\text{Inhibition Rate} = \frac{\text{RFU(test)} - \text{RFU(DMSO)}}{\text{RFU(bortezomib)} - \text{RFU(DMSO)}} \times 100\%. \quad (7)$$

In this case, fluorescence of DMSO is seen as none inhibition (0%), and fluorescence of Bortezomib is seen as completed inhibition (100%). Compounds with inhibition rates more than 20% at 20 μM are considered as hits.

**Dose-response assay and IC50 determination.** To determine $IC_{50}$, candidate inhibitors are assayed at 9 or 10 gradient concentrations.

A series of candidate inhibitor, Bortezomib, or DMSO dilutions is prepared starting from a maximum concentration of 100 μM, with each subsequent concentration being half or one third of the previous one (2-fold or 3-fold dilution gradient). $IC_{50}$ is determined by the change of recorded fluorescence (as RFU) and gradient dilution of inhibitors concentration. Non-linear fit (log(inhibitor) vs. normalized response) is used for $IC_{50}$ curve fitting.

#### Compound generation in Design and Refine stages for ClpP

**Compound generation.** Given a complex crystal structure with a protein receptor and a ligand, the center of the ligand is denoted as $c$. For each residue $i$ of a protein, if its centroid $p_i$ satisfies the condition $\|c - p_i\| \le \tau$, i.e., within a distance cutoff $\tau$ from the ligand center $c$, then residue $i$ is included in the pocket, where the distance cutoff $\tau$ is predefined.

In the case of ClpP complex, we first designed compounds based on published complex structure (PDB ID 5DZK). We took two values of $\tau$ to be 10 Å and 15 Å. Multiple binding sites can be extracted. We used beam search with beam size 20 to generate compounds. The $\beta$ of the VAE was set to be 0.1 or 1. We initialized compound generation with 20 unique random seeds, ranging from 1 to 20. After removing duplicate and invalid generated compounds, we obtained 2.6k unique compounds.

During the following Refine stage, in addition to the binding pocket information, we included guiding information encoded in 4 representative compounds and 3 experimentally discovered compounds exhibiting weak inhibition activities. The parameter $\tau$ was set to 10 Å, 12 Å, and 15 Å. We used beam search with beam sizes of 4, 10, and 20 for compound generation. The $\beta$ parameter of the VAE was set to 0.1 or 1. We initiated compound generation with 100 unique random seeds, ranging from 1 to 100. After removing duplicates and invalid compounds, we obtained a total of 8.4k unique compounds.

**UMAP visualization.** Compounds are converted to 1024-dimensional vectors with function `GetMorganFingerprintAsBitVect` from `rdkit`. UMAP transformation[70] is performed with parameters: `n_neighbors=20, min_dist=0.7, metric=jaccard`.

#### Ligand docking to protein target

The SMILES of generated compounds were converted to 3D structures with Open Babel program. Subsequently, AutoDock Tools was employed to add hydrogens and assign the Gasteiger charge to both the converted 3D compounds and the RCSB downloaded protein 5DZK before the docking process. The 5DZK ligand-centered maps were defined by the program AutoGrid and grid box was generated with definitions of 20 × 20 × 20 points and 1 Å spacing. Molecular docking was performed with AutoDock Vina program with default settings. The predicted binding poses were visualized using the PyMol program.

#### Pocket-ligand pair shuffling experiment

In this experiment, we randomized the (pocket, ligand) pairs and re-trained a TamGen model, denoted as TamGen-r. Specifically, given the training dataset $\mathcal{D} = \{(\mathbf{x}_i, \mathbf{y}_i)\}_{i=1}^N$, we shuffled the indices $I = \{1, 2, \cdots, N\}$ using the python code `random.shuffle(I)` and got the shuffled indices $\{\varsigma(1), \varsigma(2), \cdots, \varsigma(N)\}$. Consequently, we obtained a randomized dataset $\tilde{\mathcal{D}} = \{(\mathbf{x}_{\varsigma(i)}, \mathbf{y}_i)\}_{i=1}^N$. We then followed the same training and inference procedures as standard TamGen for TamGen-r.

#### Ablation studies on self-attention layers

The TamGen variant without the distance-aware attention (denoted as TamGen w/o `dist_attn`): the third equation in Eq. (2) is replaced with

$$\hat{\alpha}_j = h_i^{(l)\top} W h_j^{(l)}, \qquad (8)$$

The TamGen variant without the data augmentation from the coordiates (denoted as TamGen w/o `coord_aug`): replace Eq. (1) with

$$h_i^{(0)} = E_a a_i + E_r \left( r_i - \frac{1}{N} \sum_{j=1}^N r_j \right), \qquad (9)$$

#### Criteria for additional compounds selection

The selection criteria for the additional 8 compounds were as follows:
1. The compound pool was composed of the 296 compounds that met the criteria established by our docking results and phenotypic AI model filters.
2. These 296 compounds were clustered using MCS-based clustering with the StarDrop software, using an outlier cutoff threshold of 0.5. This process resulted in 34 clusters and 13 outliers.
3. We selected the top 10% of compounds from each cluster with better docking scores.
4. The shortlisted compounds were subject to a manual review step to assess their synthetic feasibility. Ultimately, 8 compounds were selected for synthesis.

#### Reporting summary

Further information on research design is available in the Nature Portfolio Reporting Summary linked to this article.

### Data availability

The essential data supporting the main findings of this study are available within the article, its Supplementary Information files and Zenodo (https://doi.org/10.5281/zenodo.13751391)[71]. The training and test data used in this research can be generated using the code available at https://github.com/SigmaGenX/TamGen/tree/main/data. Pre-training data can be downloaded from PubChem (https://ftp.ncbi.nlm.nih.gov/pubchem/Compound/). Source data are provided with this paper.

### Code availability

Our code is available at https://github.com/SigmaGenX/TamGen(recommended) and Zenodo (https://doi.org/10.5281/zenodo.13751391)[71]. The model weights are available via Zenodo at https://doi.org/10.5281/zenodo.13751391.

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

## Acknowledgements

We are grateful to Dr. Yang Fan for his contribution to the TamGen code. This work was done when Kehan Wu and Qizhi Pei were interns at Microsoft Research AI for Science. We thank Dr. Nathan Baker, Dr. Christopher M. Bishop, Dr. Sheng Ding, Dr. Marwin Segler, Dr. Ryota Tomioka and Dr. Rumin Zhang for their insightful discussions and feedback. This work is partially supported by Bill & Melinda Gates Foundation.

## Author contributions

Y.X. and K.W. proposed the original idea. K.W. implemented the code and processed the data. Y.X., Q.P., L.W. and S.X. diagnosized and improved the system. J.G., R.L. and H.G. designed the evaluation and filtration pipeline of TB. R.L. and Si.C. conducted the synthetic experiments. Y.C. and J.W. conducted biological experiments. Y.Z., H.G., X.L., S.H., R.L., Y.C., J.G., P.D., Y.X., K.W. and H.L. analyzed the data and results. C.-K.C., Shawn C., L.Z., N.Y., E.C., T.-Y.L. provided consultation and technical support. P.D. and H.L. led the paper writing and all authors contributed. Y.X., J.G. and T.Q. led the project.

## Competing interests

The authors declare no competing interests.
