## [Peer Review File · Nature Communications]

TamGen: Drug Design with Target-aware Molecule Generation through a Chemical Language ModelReviewer #1 (Remarks to the Author):

Summary

This paper develops a GPT-like chemical language model, TamGen, for target-aware molecule generation. Compared to previous approaches, TamGen focuses specifically on how well the generated compounds meet the physicochemical properties required for drug-like compounds, such as synthetic accessibility. Finally, the authors develop a Design-Refine-Test pipeline for drug discovery and identify seven compounds showing compelling inhibitory activity against tuberculosis.

Strength

(S1) The topic, DL-based molecular generation, has a significant potential impact on the field of drug discovery.

(S2) The proposed TamGen is pre-trained with more than 10 million compound data, from which the mined molecular semantics help molecular generation.

(S3) A case study targeting tuberculosis is provided, demonstrating the potential of TamGen for real-world biological applications.

Weakness

(W1) It has not been clearly explained whether TamGen can generate the 3D structures of molecules. It would be better to make this clearer in the Introduction and Figure 1.

(W2) There is much work related to the target-bind molecule generation, e.g., D3FG [1], DiffSBDD [2], GraphBP [3]. Please explain in detail the differences between TamGen and those.

(W3) Limitations of TamGen have not been thoroughly discussed. Please add a detailed discussion about the limitations.

(W4) Explanations for Eq. (6) are lacking, e.g. why impose a KL divergence constraint? Is $p(z)$ a prior distribution? Why is $p(z)$ set to a standard Gaussian distribution and how it would affect the method?

(W5) The proposed method was tested only on tuberculosis and it is not clear how TamGen is generalizable to other biological scenarios. The authors are encouraged to discuss some of the potential applications.

[1] Lin H, Huang Y, Zhang O, et al. Functional-group-based diffusion for pocket-specific molecule generation and elaboration. NeurIPS, 2024.

[2] Schneuing A, Du Y, Harris C, et al. Structure-based drug design with equivariant diffusion models. arXiv preprint arXiv:2210.13695, 2022.

[3] Liu M, Luo Y, et al. Generating 3d molecules for target protein binding. ICML, 2022.

Reviewer #1 (Remarks on code availability):

Remarks on code availability

I have reviewed the code released on Github, which can reproduce the paper's results. Furthermore, the authors have provided implementation details in README.md, including required packages, installations, and running scripts with rich annotations.

Reviewer #2 (Remarks to the Author):

The authors present a study that proposes an interesting approach to generate new compounds targeting specific proteins. They use a GPT-like architecture to generate molecules and combine it with cross-attention to introduce target-awareness, resulting in structures with improved properties. Additionally, the authors apply the Design-Refine-Test pipeline to identify and test ClpP inhibitors, resulting in the identification of 7 compelling inhibitors. The work is highly interesting in principle but the methodology as well as the experimental outcome are not convincing.

Major Issues

1. I would like to comment on the selection of compounds for laboratory testing. I think it is problematic to start from 296 generated compounds and look for analogues in a commercial library when the similarity score is only more than 0.55. The study seems inconsistent as only three out of the 167 tested compounds are actually novel, while the rest are commercially available analogues of the predictions. I also find it problematic to explain why the originally proposed structure (Syn-A003-01) has a higher IC₅₀ than its analogue. Furthermore, it looks like the selected compounds are very similar to the library and not in the cluster of the generated structures (Fig. 2).

a. To further validate the tool, I would suggest synthesising up to 25 molecules and validating their activity. Otherwise, no real conclusion can be drawn about the predictive capabilities of TamGen. Please describe the selection procedure, to enable comparisons between the analogous and the generated compounds.

b. In addition, I would like to see a visualisation of the similarity scores of all 159 selected compounds compared to the predicted ones.

c. It would be interesting to see the mean top-3/top-10 similarity scores of the 159 selected compound compared to the (i) training set, (ii) the commercial library and (iii) Drug Repurposing Hub with FDA-approved drugs.

d. Also, the mean similarity to (i), (ii) and (iii) could be provided for the generated (296) and non-selected generated structures (8069).

e. Please conduct a similar analysis to (c) and (d) but look at the structural diversity within the different selections instead.

f. Please provide appropriate statistical testing.

g. It is not well explained why 3 additional compounds with rather low activity against Mtb ClpP (100-200 μ M) were selected for the model.

h. To justify the "application for TB drug design" it is required to test the most active molecule(s) against mycobacterial growth.

Minor Issues

2. The experimental data is missing.

a. Show all structures and experimental data in a condensed but informative way in the SI.

3. I think the Design-Refine-Test strategy is a good way to show how to apply your approach.

However, I would like to point out the limitations of this approach. You need a known target and its structure, and you need to identify the binding site of a known interactor.

a. It should be noted that TamGen can be used if all these data are available. However, this also limits the practical use of this tool.

4. You introduced ablation studies in the Discussion and Conclusions part.

a. I would recommend mentioning them early and explain why they were selected and how they are designed.

5. I would also like to mention Fig. 4. In my opinion, the UMAP does not seem to cluster correctly, probably due to the large number of compounds. Also, the structures are not easy to observe, and the aromaticity is represented inconsistently.

a. Please test other parameters of the UMAP to improve the clustering.

b. Furthermore, please address the readability and consistency of the figure.

6. I noticed that Fig. S2 shows four second or third ranks for TamGen (Docking, QED, SAS, Mol. Diversity) and perhaps the first rank with the Lipinski rules.

a. Are these results statistically significant and why is the ranking different from Fig. 2 and Tab. S1 in the main text?

b. I think showing score and rank at the same time, makes the figure hard to understand.

Reviewer #3 (Remarks to the Author):

This manuscript introduces TamGen, a transformer-based language model designed for generating target-aware molecules that exhibit desired drug-related characteristics. It employs geometric self-attention to capture the spatial information of proteins and integrates this with the latent vectors of seed molecules, which are learned through a Variational Autoencoder (VAE). This fused information is subsequently fed into the transformer decoder to learn the joint distribution of molecules and proteins. Consequently, TamGen possesses the capability to produce molecules that are both influenced by the target protein and centered around the seed molecules. Furthermore, the authors undertook wet lab experiments, i.e., generating molecules targeting the Tuberculosis ClpP protease and computing their IC₅₀ values. The authors adopted the Design-Refine-Test workflow, enabling iterative improvements of the generated molecules and thereby highlighting TamGen's effectiveness in generating viable drug candidates.

This study is interesting and offers a promising approach to drug design. This paper is clearly written and I can fully follow up their works. However, there are a few concerns that need to be addressed.

1、 The benchmark is somewhat unbalanced. All methods in the benchmark generate

molecules in 3D Euclidean space while TamGen generates molecules in 1D SMILES sequence. Different molecular representations come with their own strengths and weaknesses. 3D representations excel at capturing the interactions between proteins and ligands, whereas 1D representations are adept at generating molecules with high validity and sensible structures, like fewer fused rings. Therefore, for a more balanced comparison, the authors should also consider comparing their methods with 1D SMILES-based models, as exemplified by references [1-2].

[1] Transformer neural network for protein-specific de novo drug generation as a machine translation problem. *Sci Rep*.

[2] AlphaDrug: protein target specific de novo molecular generation, *PNAS Nexus*, Volume 1, Issue 4, September 2022, pgac227.

Besides, more recently published methods [3-6] should be introduced in the main body.

[3] Generation of 3D molecules in pockets via a language model. *Nat Mach Intell* 6, 62–73 (2024).

[4] KGDiff: towards explainable target-aware molecule generation with knowledge guidance, *Briefings in Bioinformatics*, Volume 25, Issue 1, January 2024, bbad435.

[5] PocketFlow is a data-and-knowledge-driven structure-based molecular generative model. *Nat Mach Intell* 6, 326–337 (2024).

[6] 3D molecular generative framework for interaction-guided drug design. *Nat Commun* 15, 2688 (2024).

2、 Many deep learning models, such as [7-9], are SE(3)-equivariant or SE(3)-invariant with carefully designed constrains on model structures. However, TamGen simply learns geometric protein information with data augmentation to the coordinates and a distance-aware attention, while achieves promising performance in wet experiments. It would be better if the authors can provide an ablation study to figure out the effectiveness of the two techniques.

[7] E (n) equivariant graph neural networks[C]//International conference on machine learning. PMLR, 2021: 9323-9332.

[8] SE (3) equivariant graph neural networks with complete local frames[C]//International Conference on Machine Learning. PMLR, 2022: 5583-5608.

[9] Diffpack: A torsional diffusion model for autoregressive protein side-chain packing[J]. *Advances in Neural Information Processing Systems*, 2024, 36.

3、 This question serves as a follow-up. Almost all methods in the benchmark pointed out that the 3D Euclidean interactions between target proteins and ligands are crucial for ligands to successfully bind to their respective target proteins. However, TamGen does not model any above 3D interactions while still achieves good performance in wet experiments. It seems the seeding compounds in Design-Refine-Test pipeline play the key role. If feasible, it would be insightful if the authors could conduct wet lab experiments using molecules generated by other methods that explicitly account for 3D interactions, such as TargetDiff, for comparison. It would be highly significant if the authors could show that explicitly modeling the 3D interactions between proteins and ligands might not be as crucial, provided that the generated compounds can undergo iterative improvements through the Design-Refine-Test pipeline.

4、 The capability of TamGen to accurately capture the context of the protein pocket remains unclear. It would be insightful to explore whether TamGen can maintain its performance on the MRR metric across protein pockets of varying sizes. Additionally, it's worth investigating if TamGen has the ability to generate molecules that are sensitive to different protein isoforms.

5、 Using Mean Reciprocal Rank (MRR) based on both Lipinski's rule of five and logP might introduce a bit of overlap, as Lipinski's rule already takes logP into consideration among its criteria.

6、 minor changes:

a) Figure S4 could be enhanced by adjusting the line width and font size for clearer visualization.

b) In Figure 4a, it would be beneficial to standardize the representation of benzene rings for clarity and consistency.

Reviewer #4 (Remarks to the Author):

The authors introduce TamGen, a target-aware language model for generating potential drug-like molecules. They demonstrate the effectivity of their model by comparison with others similar, and implement a drug-design pipeline using their model where they in the end obtain a few candidates with experimentally validated outstanding properties.

The work offers a novel approach to generating molecules in a targeted manner, and provide evidence of its effectiveness.

Some of the claims are potentially not well supported by evidence, and would require either further experiments or rephrasing.

The authors perform some ablation study, however the work can heavily benefit from more in-depth studies, in particular I would propose performing the following:

- randomize the (pocket, ligand) pairs during training, and reproduce Fig2b. That way you'll prove that the model is actually leveraging information from the (pocket, ligand) pairs instead of some spurious correlation.

- The authors claim that the GPT model can leverage few data efficiently. Could you assess the effect of the model type to support this claim? e.g. replace the GPT part by an RNN.

Other experiments, potentially future work:

- The authors investigate the binding mechanisms of some of the generated and experimentally validated compounds. Could you prove the model is somehow aware of these connections and learns to find them? Or what is the explanation for the better docking scores overall? Maybe visualizing attention maps could help this.

Appart from that, fix this typo in the manuscript:

line 454: change "urine" to "urea"

In Figures 2d and 4a, there is an inconsistent use of kekulized structures along with resonance structures (explicit double bonds in aromatic ring). I would advice to select one and stick to it for publication.

Responses to comments by Reviewer #1

This paper develops a GPT-like chemical language model, TamGen, for target-aware molecule generation. Compared to previous approaches, TamGen focuses specifically on how well the generated compounds meet the physicochemical properties required for drug-like compounds, such as synthetic accessibility. Finally, the authors develop a Design-Refine-Test pipeline for drug discovery and identify seven compounds showing compelling inhibitory activity against tuberculosis.

Strengths:

(S1) The topic, DL-based molecular generation, has a significant potential impact on the field of drug discovery.

(S2) The proposed TamGen is pre-trained with more than 10 million compound data, from which the mined molecular semantics help molecular generation.

(S3) A case study targeting tuberculosis is provided, demonstrating the potential of TamGen for real-world biological applications.

Reply: We appreciate your expert opinions and valuable feedback on our manuscript. In accordance with your insightful suggestions, we have made the following clarifications and enhancements:

1. Clarified the uniqueness and connections between TamGen and other state-of-the-art approaches for molecules generation.
2. Provided a detailed discussion on the applications and limitations of TamGen.
3. Explained in detail the parameters used in Equation (6) of the training loss, addressing specific concerns regarding the KL divergence constraint and the choice of a standard Gaussian distribution as the prior.

We hope that our responses, including additional model testing, wet lab experiments, and revision, will address raised concerns and enhance the quality of our work.

Question 1: It has not been clearly explained whether TamGen can generate the 3D structures of molecules. It would be better to make this clearer in the Introduction and Figure 1.

Reply: Thanks for pointing out this issue. TamGen is designed to generate 1D SMILES representations of small molecules, rather than providing 3D structures directly. We added clear statements related to this in the Introduction (Lines 91-94, highlighted in red) and in the caption of Figure 1 (Lines 172, 180, highlighted in red).

Question 2: There is much work related to the target-bind molecule generation, e.g., D3FG [1], DiffSBDD [2], GraphBP [3]. Please explain in detail the differences between TamGen and those.

047 [ref1] Lin H,Huang Y, Zhang O, et al. Functional-group-based diffusion for
048 pocket-specific molecule generation and elaboration. NeurIPS, 2024.

049 [ref2] Schneuing A, Du Y, Harris C, et al. Structure-based drug design with
050 equivariant diffusion models. arXiv preprint arXiv:2210.13695, 2022.

051 [ref3] Liu M,Luo Y, et al. Generating 3d molecules for target protein
052 binding. ICML, 2022.

053 **Reply:** Thank you for highlighting these related works. All these methods are
054 developed for the molecular generation in 3D space. These methods can be
055 categorized as either autoregressive or diffusion-based approaches. We added
056 the method description and comparison to the revised manuscript (references
057 are cited in Lines 74-76 and detailed discussion in Lines 633-648). Here we also
058 provide a brief summary:

- 059 1. D3FG [ref1]:This approach introduces a novel decomposition of molecular
060 generation into two stages: functional group creation and linker assembly.
061 Diffusion models are applied to both stages, facilitating a structured and
062 comprehensive generation process.
- 063 2. DiffSBDD [ref2]: This paper and TargetDiff [1] share a similar idea of lever-
064 aging diffusion models to generate compounds in 3D space. In terms of atom
065 type generation, TargetDiff uses discrete diffusion (i.e., a categorical distri-
066 bution to model discrete atom types), while DiffSBDD chooses to diffuse
067 the atom features in continuous space.
- 068 3. GraphBP [ref3]: An example of an autoregressive method, GraphBP gener-
069 ates both atomic coordinates and atom types sequentially. Pocket2Mol [2]
070 follows a similar approach but with a different network backbone.

072 Differences with TamGen:

- 073 • Dimensionality: TamGen focuses on 1D generation using SMILES, as
074 opposed to the 3D space considered in the aforementioned methods.
- 075 • Refinement Capability: TamGen has a unique capability to refine com-
076 pounds within the latent space of a neural network, which is not explicitly
077 addressed in the other works.

079 We acknowledge that 3D interactions between compounds and proteins are
080 critical for structure-based drug design. 1D generation methods like TamGen
081 might not fully leverage the intricate interactions available in geometric space.
082 In future work, we plan to integrate insights from 3D generation methods,
083 specifically the autoregressive approaches, to enhance the performance of Tam-
084 Gen. This integration aims to improve compound generation and optimization,
085 with benefit of improving 3D structure quality and drug-target interactions.

087 **Question 3:** Limitations of TamGen have not been thoroughly discussed.
088 Please add a detailed discussion about the limitations.

089 **Reply:** We appreciate the constructive suggestions. To address this concern,
090 we have added dedicated paragraphs in the discussion section (Line 624 to
091

Line 632, highlighted in red). Below is a summary of the limitations in the current implementation of TamGen:

1. Sensitivity to Protein Sequence Differences: Although the chemical properties are encoded in the protein sequences, the current TamGen does not accurately distinguish targets with small variations, such as point mutations or different protein isoforms that do not directly affect the binding pocket features. The low sensitivity to protein sequences may reduce TamGen's effectiveness in certain specific applications, such as selectivity of generated compounds against homologous targets with similar pockets.
2. Dependence on Structural Information: As TamGen employs a structure-based drug design approach, it relies on the availability of target protein structures and detailed binding pocket information. This dependence restricts its applicability in scenarios where such structural data is unavailable or incomplete. Predicted 3D structures can help fill this void, but the performance is to be tested.
3. 1D vs. 3D Generation: Currently, TamGen utilizes 1D generation using SMILES representation. This approach may not capture all important interactions present in three-dimensional structural space. Future development can benefit from incorporating 3D generation methodologies to enhance compound generation and optimization processes.
4. Property Optimization: TamGen currently does not directly optimize binding affinities or other critical properties of compounds. The generation process can be coupled with reinforcement learning or other techniques with guiding signals for stronger binding and drug properties.

On the basis of the current model, these limitations can be overcome in future development to enhance the performance and applicability of TamGen. With these additional discussions of limitations, the present work also lays out directions for future development. Thank you for bringing this to our attention.

Question 4: Explanations for Eq. (6) are lacking, e.g. why impose a KL divergence constraint? Is $p(z)$ a prior distribution? Why is $p(z)$ set to a standard Gaussian distribution and how it would affect the method?

Reply: Thanks for your comments. $p(z)$ is a prior distribution as you recognized. It is a common practice to set $p(z)$ to be a standard Gaussian distribution due to its convenient mathematical properties, particularly in the calculation of the KL divergence. Imposing a KL divergence constraint ensures that the learned latent space resembles the prior distribution $p(z)$, which promotes smooth interpolation and generalization. It regularizes the encoder by pulling the approximate posterior distribution (the encoder's output) closer to the prior distribution (typically a standard Gaussian). This design not only ensures meaningful latent representations, but also helps prevent overfitting. We add the revisions to Line 776-784.

**Question 5:** The proposed method was tested only on tuberculosis and it
is not clear how TamGen is generalizable to other biological scenarios. The
authors are encouraged to discuss some of the potential applications.

**Reply:** Thank you for this valuable suggestion. We fully agree that the gen-
eral application of TamGen requires more real world evidences. In this work,
we present two types of evidences: one is on the benchmark dataset analy-
sis based on existing experimental or computational data; and the other is
to design compound molecules for targets that are under investigation. Due
to constraints in time and resources, we focused our efforts on two protein
targets that are associated with the most infectious diseases: (1) the ClpP
protein of Mycobacterium tuberculosis; and (2) a well-characterized protease
target of SARS-CoV-2. While we have provided a comprehensive account
of the proposed molecules and experimental processes for TamGen-based
ClpP inhibitor discovery, we have also identified promising hit compounds
for the SARS-CoV-2 protein target using a similar pipeline. However, due to
some policy restrictions, the specifics of the SARS-CoV-2 compounds can-
not be disclosed in this manuscript. Related information can be found at:
[https://www.microsoft.com/en-us/research/blog/ghddi-and-microsoft-re-](https://www.microsoft.com/en-us/research/blog/ghddi-and-microsoft-research-use-ai-technology-to-achieve-significant-progress-in-discovering-new-drugs-to-treat-global-infectious-diseases/)
[search-use-ai-technology-to-achieve-significant-progress-in-discovering-new-](https://www.microsoft.com/en-us/research/blog/ghddi-and-microsoft-research-use-ai-technology-to-achieve-significant-progress-in-discovering-new-drugs-to-treat-global-infectious-diseases/)
[-drugs-to-treat-global-infectious-diseases/](https://www.microsoft.com/en-us/research/blog/ghddi-and-microsoft-research-use-ai-technology-to-achieve-significant-progress-in-discovering-new-drugs-to-treat-global-infectious-diseases/). We hope that the capability of
TamGen is sufficiently demonstrated by the benchmark analysis along with
comparison with other generation methods, as well as the real world evidence
on ClpP inhibitor design.

As highlighted by the reviewer, TamGen is a versatile tool for compound
generation, broadly applicable to various scenarios in structure-based drug
design. It can generate novel compounds, as well as further optimization, using
only the structural and pocket information of target proteins, making it a user-
friendly tool. We have incorporated the suggested discussions into the revised
manuscript (Line 663 - Line 670, highlighted in red).

References

[1] Jiaqi Guan, Wesley Wei Qian, Xingang Peng, Yufeng Su, Jian Peng, and
Jianzhu Ma. 3d equivariant diffusion for target-aware molecule generation
and affinity prediction. *The Eleventh International Conference on Learning*
*Representations*, 2023.

[2] Xingang Peng, Shitong Luo, Jiaqi Guan, Qi Xie, Jian Peng, and Jianzhu
177 Ma. Pocket2mol: Efficient molecular sampling based on 3d protein pockets.
*International Conference on Machine Learning*, 2022.

Responses to comments by Reviewer #2 185

The authors present a study that proposes an interesting approach to generate 186
new compounds targeting specific proteins. They use a GPT-like architec- 187
ture to generate molecules and combine it with cross-attention to introduce 188
target-awareness, resulting in structures with improved properties. Addition- 189
ally, the authors apply the Design-Refine-Test pipeline to identify and test 190
ClpP inhibitors, resulting in the identification of 7 compelling inhibitors. The 191
work is highly interesting in principle but the methodology as well as the 192
experimental outcome are not convincing. 193

Major Issues: I would like to comment on the selection of compounds for 194
laboratory testing. I think it is problematic to start from 296 generated com- 195
pounds and look for analogues in a commercial library when the similarity 196
score is only more than 0.55. The study seems inconsistent as only three out of 197
the 167 tested compounds are actually novel, while the rest are commercially 198
available analogues of the predictions. I also find it problematic to explain why 199
the originally proposed structure (Syn-A003-01) has a higher IC₅₀ than its 200
analogue. Furthermore, it looks like the selected compounds are very similar 201
to the library and not in the cluster of the generated structures (Fig. 2). 202

Reply: Thank you for recognizing the potential and interest in our study. 203
We appreciate your insightful comments regarding the selection of compounds 204
for laboratory testing. We understand your concerns about the consistency 205
of our approach and the selection criteria based on similarity scores. Before 206
addressing your concerns, we hope to clarify the application of TamGen and 207
the rationale behind our approach. While TamGen has the capability to gen- 208
erate novel compounds, the direct utility of these compounds in their original 209
form can be limited, due to a lack of comprehensive in vivo property data, such 210
as toxicity, metabolism, and pharmacokinetics. More critically, the time and 211
resources required to synthesize novel compounds can be a bottleneck to con- 212
duct large scale experimental testing. Therefore, we leverage the commercially 213
available compounds for the application case of ClpP inhibitor design. On the 214
other hand, we agree with you that novelty of compounds is of high interest 215
to both methodology developers and users. Therefore, we synthesized sev- 216
eral generated compounds and conducted experiments on them. The ultimate 217
goal of TamGen is to help discover molecules that achieve desired functions, 218
so we focus on the actual effectiveness instead of whether the molecules are 219
completely novel or analogs. 220

With the above rationale, we introduced a library search step as part of 221
our Design-Refine-Test pipeline that combines the advantages of de novo drug 222
generation and traditional screening methods. We searched for analogues in 223
commercially available libraries using a relatively loose similarity threshold 224
of 0.55. This approach allowed us to test the biological activity of readily 225
available compounds that are structurally related to the generated ones. This 226
approach enabled fast identification of promising candidates and gained prelim- 227
inary insights into their activity profiles. Following this fast turn-around initial 228

screening, we proceeded to synthesize the generated compounds and conduct
a detailed structure-activity relationship (SAR) analysis. This iterative pro-
cess leverages the structural similarities between the generated compounds and
their analogues, facilitating a more efficient optimization of lead compounds.
This hybrid approach, combining AI-generated diversity with accessible ana-
logue testing, exploits the benefits of both methodologies and accelerates the
drug discovery process.

Regarding the small number of novel compounds being experimentally
tested, we hope to emphasize that the primary goal of TamGen is to provide
a diverse and promising compound pool to help discover and optimize com-
pounds for given targets. The focus on analogues (which are discovered via the
generated compounds by TamGen) in the initial testing phase is a strategic
decision to expedite the identification of active compounds and gather crucial
SAR data, providing valuable information for subsequent rounds of compound
refinement.

As for the specific case of the originally proposed structure (Syn-A003-
01) having a higher IC₅₀ than its analogue, this outcome underscores the
importance of SAR analysis. Variations in chemical structure, even subtle
ones, can significantly impact biological activity. The analogue may exhibit
improved binding affinity or better pharmacodynamic properties, which are
critical insights gained through our iterative testing approach.

We have carefully reviewed our manuscript to ensure that we do not over-
claim the immediate utility of the generated molecules. We have explicitly
discussed potential knowledge gaps between generation methods and library
search actions, and outlined potential future improvements (Line 607 to Line
623).

In summary, our approach aims to balance the novelty and potential of
AI-generated compounds with the practicality and speed of analogue testing,
thereby streamlining the drug discovery pipeline. We believe this strategy effec-
tively harnesses the strengths of both methodologies to identify and optimize
promising drug candidates.

Thank you again for your valuable feedback.

**Question 1a:** To further validate the tool, I would suggest synthesising up
to 25 molecules and validating their activity. Otherwise, no real conclusion
can be drawn about the predictive capabilities of TamGen. Please describe
the selection procedure, to enable comparisons between the analogous and the
generated compounds.

**Reply:** We appreciate the reviewer's suggestion to demonstrate the effec-
tiveness of TamGen by validating the activities of generated compounds. As
mentioned, molecular synthesis, especially for novel compounds, is a limiting
step, because of small throughput and high demands in resources. Despite these
limitations, we managed to synthesize 8 novel compounds and verified their
enzymatic inhibition activity. The selection procedure for these 8 compounds
is described in the following:

1. The compound pool was composed of the 296 compounds that met the criteria established by our docking results and phenotypic AI model filters.
2. These 296 compounds were clustered using MCS-based clustering with the StarDrop software, using an outlier cutoff threshold of 0.5. This process resulted in 34 clusters and 13 outliers.
3. We selected the top 10% of compounds from each cluster with better docking scores.
4. The shortlisted compounds were subject to a manual review step to assess their synthetic feasibility. Ultimately, 8 compounds were selected for synthesis.

Fig. R1: The novel compounds generated by TamGen and their IC₅₀ values measured in this study.

As shown in Figure R1, six of the synthesized compounds exhibited IC₅₀ values below 40 μM . These results further validate our current approach, which is effective in generating promising hit compounds for the ClpP protein target.

We have included these experimental details into the manuscript to better demonstrate the capability of our model (Line 967 to Line 979 for the selection criteria, and Line 565 to Line 570 for results). Thank you once again for your suggestion.

Question 1b: In addition, I would like to see a visualisation of the similarity scores of all 159 selected compounds compared to the predicted ones.

Reply: In Figure S8 of the supplementary information (also illustrated in Fig. R2), we provided a visualization of the similarity scores between generated compounds and the analog compounds. Specifically, the x-axis represents the Maximum Common Substructure (MCS) scores, which indicate the structural similarity between the analogue compounds and the generated compounds. The y-axis shows the single-dose inhibition rates. The figure shows that most MCS scores are in the range of 0.6 to 0.7. The average value of the 159 MCS scores is 0.681, associated with a standard deviation of 0.090. Also, to be mentioned, the similarity score is not significantly correlated with the inhibition

rated quantified by single dose assays (Spearman correlation = 0.158). There-
 fore, from a retrospective perspective, selecting a wider and not so rigorous
 similarity score might offer better opportunities to find good candidate. We
 have updated the figure caption to include this information for clarity.

**Fig. R2: Inhibition rate of the 159 library search analogs relative to**
 **Bortezomib.** All compounds were evaluated at the concentration of 20 μ M.
 The dashed line indicates the threshold for analog selection. *x*-axis: Maximum
 Common Substructure (MCS) similarity scores. **The mean values and standard**
 **derivations of the MCS scores of the 159 selected compounds are 0.681 and**
 **0.090, respectively. The Spearman correlation between MCS similarity scores**
 **and inhibition rates is 0.158.** See Methods for details.

**Question 1c to 1f:**

[c] It would be interesting to see the mean top-3/top-10 similarity scores of
 the 159 selected compound compared to the (i) training set, (ii) the commercial
 library and (iii) Drug Repurposing Hub with FDA-approved drugs.

[d] Also, the mean similarity to (i), (ii) and (iii) could be provided for the
 generated (296) and non-selected generated structures (8069).

[e] Please conduct a similar analysis to (c) and (d) but look at the structural
 diversity within the different selections instead.

[f] Please provide appropriate statistical testing. (for the above figures)

Reply: We thank the reviewer for the constructive suggestions and detailed instructions. To verify the similarity between our generated compounds and their corresponding analogue compounds in established databases, we computed the similarity scores of 296 generated compounds, 159 analogue compounds (denoted as “original” and “analogue”, respectively) against three datasets: (1) the dataset used for training our model (briefly denoted as “training set”); (2) the Drug Repurposing Hub with FDA-approved drugs (denoted as “FDA approved”); (3) a randomly selected subset of 1 million compounds from Enamine, which is a commercial compound library (denoted as “Enamine”). We analyzed the results in all cases for both the top-3 and top-10 similarity metrics.

More specifically, for each ligand l , from either generated set or analog set, we computed the Morgan fingerprint similarity between l and every ligand in the reference datasets (training set, FDA approved, or Enamine). Subsequently, we calculated the average of the top-3 and top-10 fingerprint similarity scores. Finally, the mean scores are visualized by using violin plot in Figure R2(a,b), with the p -values shown in Figure R2d.

During the analysis of compound similarity, we observed the following:

1. For both top-3 or top-10 similarity metrics, the generated compounds exhibit the highest similarity to the training set. This outcome is expected since a machine learning model is designed to capture the distribution characteristics of the data. The similarity to Enamine, the commercial chemical library, was ranked to be the second. This correlation is logical given the extensive size and diversity of the commercial library, which encompasses a broad spectrum of potentially beneficial compounds.
2. In terms of top-3 and top-10 similarity, the analogs display the highest similarity to the commercial library Enamine. This is attributable to the fact that the high-throughput screening (HTS) library used in our experiments is encompassed within the large commercial library. In the ideal case, the analogs can be found in the commercial library, yielding very high similarity score if considering only the most similar matches. In this analysis, the similarity scores did not reach the maximum value of 1 due to our utilization of only a subset of Enamine (1 million compounds) to manage computational costs. Nonetheless, the observed pattern is clear and logical.

We additionally present the structural diversity among the 296 chosen compounds, the 159 analog compounds, and the remaining pool of 8,069 compounds. The structural diversity between any two compounds c_1 and c_2 is defined as “ $1 - \text{fingerprint_similarity}(c_1, c_2)$ ”. This calculation of structural diversity is performed for each pair of distinct compounds within their respective groups (i.e., the 296 chosen compounds, the 159 analog compounds, and the remaining pool of 8,069 compounds). The outcomes of this analysis are in Figure R2(c). From the data, we can see that the set of 296 selected compounds exhibits the highest level of structural diversity, succeeded by the analog compounds. This observation underscores the efficiency of generative techniques

(a) Fingerprint similarity of the selected 296 compounds and their 159 analog compounds to the training set, FDA-approved drugs, and a subset of the Enamine library.

(b) Fingerprint similarity of the remaining 8069 compounds to the training set, FDA-approved Drugs, and a subset of the Enamine library.

(c) Pairwise diversity of the 296 selected compounds, 159 analogue compounds and the remaining 8069 compounds.

	p -values for subfigure (a)			p -values for subfigure (b)		
	id1, id2	Top-3	Top-10	id1, id2	Top-3	Top-10
Id for subfigure (a)						
1. (original, training set)						
2. (original, FDA approved)	1,2	**	**	1,2	**	**
3. (original, subset of Enamine)	1,3	**	**	1,3	**	**
4. (analogue, training set)	1,4	0.418	0.089	2,3	**	**
5. (analogue, FDA approved)	1,5	**	**			
6. (analogue, subset of Enamine)	1,6	**	**			
				p -values for subfigure (c)		
Id for subfigure (b)				id1, id2	p -value	
1. (original, training set)	2,3	**	**	1,2	**	
2. (original, FDA approved)	2,4	**	**	1,3	**	
3. (original, subset of Enamine)	2,5	**	**	2,3	**	
	2,6	**	**			
Id for subfigure (c)						
1. selected (296)	3,4	**	**			
2. analogue (159)	3,5	**	**			
3. remaining (8069)	3,6	**	**			
	4,5	**	**			
** indicates p -value < 10 ⁻⁶	4,6	**	**			
	5,6	**	**			

(d) *p*-values of the statistics from subfigure (a) to (c).

Fig. R2: Fingerprint similarity of the generated compounds and their analogue compounds to established Compound Libraries. The analysis is partitioned into the 296 compounds and their corresponding analogs (depicted in subfigure (a)) and the remaining compounds (depicted in subfigure (b)). The structure diversity of the selected 296 compounds, the 159 analogue compounds and the remaining 8069 compounds are shown in subfigure (c). (d) shows the *p*-values calculated using the Mann-Whitney *U* test between pairwise samples.

in spanning a broader expanse of chemical space. Nevertheless, it is also noted that the diversity within the selected compounds is relatively moderate, with a majority of diversity values falling below 40%.

To address the observed limitations in diversity, future work could involve refining the generative model by incorporating diversity-promoting objectives or by expanding the training dataset to include a wider variety of chemical structures. Additionally, employing diversity-oriented selection criteria during the compound selection phase could help in capturing a broader representation of the model’s generative capabilities. We appreciate the reviewer for this suggestion, so that we have the chance to identify potential issues in the current implementation.

Question 1g: It is not well explained why 3 additional compounds with rather low activity against Mtb ClpP (100-200 μ M) were selected for the model.

Reply: Thank you for your insightful query regarding the selection of three additional compounds with relatively low activity against Mtb ClpP (100-200 μ M) for our model. This decision was guided by the principles of fragment-based drug discovery [1]. In fragment-based drug discovery, the initial focus is to identify small and simple molecules known as “fragments”, which often exhibit low affinity when binding to target sites. Despite their low initial activity, these fragments offer significant advantages due to their simplicity and

507 small sizes. For example, they can serve as strategic starting points in the drug
discovery process, enabling incremental improvements through iterative cycles
of design, synthesis, and testing. These previously identified compounds have
their values in guiding the compound generation, because each must possess
certain favorable interactions with the ClpP target protein. The inclusion of
these weak inhibitors aligns well with the refinement capabilities of TamGen. In
this context, the inclusion of these additional compound information increases
the probability of generating compounds that have favorable interactions with
the target protein.

To help general readers understand the underneath rationale, we have
included the detailed descriptions about the selection of these compounds in
the Supplementary Information (Line 783 to Line 798), which are mentioned
in the main text in Line 429 - Line 432.

**Question 1h:** To justify the “application for TB drug design” it is required
to test the most active molecule(s) against mycobacterial growth.

**Reply:** We agree with the reviewer on the necessity of testing the most active
molecules against mycobacterial growth through in vivo experiments, which is
an essential step towards drug optimization and development.

In this work, our primary focus is on hit identification and optimiza-
tion. This early drug discovery phase includes generating potential compounds
and verifying their inhibitory activity against the target protein, in this
case, the Tuberculosis ClpP protease. By focusing on target engagement and
optimization through AI, we establish a robust foundation for subsequent com-
prehensive in vitro and in vivo anti-TB evaluations. This is the widely taken
strategy, securing enzyme activity inhibition as the prerequisite for in vivo
research.

While we acknowledge the importance of testing against mycobacterial
growth to further validate the efficacy of these compounds in a biological
context, due to the limited resources and time, we focus on hit identification
and keep comprehensive lead optimization and validation in vivo as follow-
up projects. Although we did not perform cell-level experiments, the results
and findings should provide sufficient support for the TamGen as an effective
method for compound generation in drug discovery research.

To be more precise on the achievement and limitation in this work, we
revised some statements by emphasizing the method rather than utilizing
the methods to fully solve problems. We have restricted our terminology to
reflect the in vitro experiments on target protein activity inhibition, using
phrases such as “inhibitory activity against the Tuberculosis ClpP protease”
instead of “inhibitory activity against Tuberculosis”. We have also reviewed
the manuscript to ensure there are no implications that cell-level activity was
tested, and we have clarified that only enzymatic level activities of the com-
pounds were mentioned. We have also explicitly stated in our manuscript
that these experiments are planned for future research to ensure a thorough
validation process (Lines 657 – 660).

We appreciate your understanding and look forward to addressing this critical aspect in subsequent studies, thereby strengthening the applicability of our findings for drug design in the real-world.

Question 2a: The experimental data is missing. a. Show all structures and experimental data in a condensed but informative way in the SI.

Reply: We appreciate the reviewer's comment. We have included the following information to address the concerns:

1. **Structures Data:** All generated compounds and their structures are now available at the following website: https://github.com/AlphaGenX/TamGen/tree/main/ClpP_inhibitor. This link is also referenced in the Data Availability section of the manuscript.
2. **Dose-Response Assay Data:** The dose-response assay data for IC₅₀ determination has been provided in a supplementary spreadsheet for easy reference and thorough analysis.
3. **Synthesis Path:** Detailed synthesis pathways for the generated compounds are included in the supplementary document to ensure clarity and reproducibility.

Question 3: I think the Design-Refine-Test strategy is a good way to show how to apply your approach. However, I would like to point out the limitations of this approach. You need a known target and its structure, and you need to identify the binding site of a known interactor. a. It should be noted that TamGen can be used if all these data are available. However, this also limits the practical use of this tool.

Reply: Thank you for your thoughtful feedback on our Design-Refine-Test strategy. We acknowledge the limitations you pointed out regarding the necessity of having a known target, its structure, and the binding site of a known interactor for effective application of the TamGen method. We have now explicitly mentioned this limitation in Lines 627 to 632 of the revised manuscript. While this requirement does pose a limitation, as highlighted by the reviewer, the recent advances in predictive tools such as AlphaFold2/3 for protein structure predictions hold promise in mitigating these concerns. A recent study [3] demonstrates that high-confidence protein structure predictions from AlphaFold2 are as effective as experimental structures for prospective docking campaigns aimed at identifying new ligands. These advancements significantly broaden the applicability of our method, even for proteins that lack experimentally determined structures. Moreover, we can leverage computational tools to identify potential binding pockets in proteins, thereby extending the practical utility of TamGen to scenarios where direct binding site information is unavailable. These approaches collectively enhance the feasibility of applying TamGen in a wider range of biological contexts. In summary, the lack of structural and binding site data may limit direct application of TamGen in drug discovery research, but the integration of emerging predictive tools and computational

techniques will facilitate the practical applications of TamGen, making it a
valuable resource in drug discovery and other related fields.

**Question 4:** You introduced ablation studies in the Discussion and Conclu-
sions part. (a). I would recommend mentioning them early and explain why
they were selected and how they are designed.

**Reply:** We appreciate your recommendation and have made adjustments to
mention the ablation studies earlier in the manuscript to explain their selection
and design. Specifically, 1) we moved the previous ablation experiment to a new
section (Line 538 to 570) in the manuscript, 2) we performed two additional
ablation experiments to understand TamGen’s performance and limitations,
and 3) we added the details of the ablation experiments to the Methods section.
In summary, we conducted ablation experiments to investigate the following
factors:

- 1. Necessity of Pre-training: We found that omitting pre-training resulted in
worse docking scores and over-simplified compound structures (Figure S9).
- 2. Effectiveness of Pocket-Ligand Pairing: Randomly shuffling pocket-ligand
pairs during training led to significant reductions in interactions between
ligand and receptor, reflected on worse docking scores (Figure S10).
- 3. Impact of Self-Attention Variant: Excluding distance-aware attention and
coordinate data augmentation notably reduced docking scores, highlighting
the importance of these techniques (Table S3).

**Question 5:** I would also like to mention Fig. 4. In my opinion, the UMAP
does not seem to cluster correctly, probably due to the large number of com-
pounds. Also, the structures are not easy to observe, and the aromaticity is
represented inconsistently. a. Please test other parameters of the UMAP to
improve the clustering. b. Furthermore, please address the readability and
consistency of the figure.

**Reply:** We thank the reviewer for the constructive suggestions and the deep
thinking. Indeed, a good visualization of the results will provide insights that
are hard to grasp through word descriptions. We conducted several testing
following the suggestions to improve the visualization of the results. First,
we reduced the dataset from 110,995 to 10,019 molecules by random sam-
pling before UMAP transformation. For quantifying the distances between the
input data in the high-dimensional space, we chose the Jaccard metric, given
that each molecule is described by a binary molecular fingerprint. Addition-
ally, to ensure a better preservation of the overall topological structure, we
adjusted the `min_dist` parameter to a higher value, which discourages the
UMAP algorithm from clustering points too closely.

To further enhance the clarity and readability of the figures, we re-organized
the information presented in both the panels and legends of the figure,
where modifications are highlighted in red. In alignment with the reviewer’s
recommendation, we also standardized the illustrations of benzene rings.

Question 6: I noticed that Fig. S2 shows four second or third ranks for Tam-Gen (Docking, QED, SAS, Mol. Diversity) and perhaps the first rank with the Lipinski rules. a. Are these results statistically significant and why is the ranking different from Fig. 2 and Tab. S1 in the main text? b. I think showing score and rank at the same time, makes the figure hard to understand.

Reply: Thanks for careful reviewing and pointing out this inconsistency to us. We are grateful to your expert opinions. In the revised version, we made the following updates:

a. Figures 2a, S2, and Table S1 are intended to present the same scores and rankings. Upon thorough review, we discovered that the x -axis labels in Figure S2 were incorrectly positioned. We have corrected the order of the bar-plots in Figure S2 in our current manuscript version. Additionally, to enhance clarity, we made explicit statements on the aggregation method (mean) used in the legends of figures and tables (highlighted in red). Furthermore, we conducted Mann-Whitney U tests to compare the TamGen method with other alternative approaches across each metric. These results are now included in Figure S2 and Table S1.

b. The practice of displaying scores and rankings within the same plot is becoming increasingly common in recent literature [2, 4], as it provides a more compact view in a single visualization. However, we recognize that distinguishing between darkness and sizes—which represent correlated but distinct types of information—may present readability challenges. To enhance clarity, we have made two adjustments: 1) We highlighted the top three ranked methods (No. 1, 2, and 3) within the plot itself, and 2) We revised the legend for Figure 2a to better explain the visual encoding used. Specifically, the size of the dots is directly proportional to the scores, whereas the darkness indicates the ranking of each method within the given metric. Scores (median) were normalized from 0% to 100% for each metric, with absolute values applied for the normalization of docking scores.

These changes aim to improve the clarity and comprehensibility of our visual representations, ensuring that our findings are communicated clearly.

References

- [1] Philine Kirsch, Alwin M Hartman, Anna K H Hirsch, and Martin Empting. Concepts and core principles of fragment-based drug design. *Molecules*, 24(23):4309, November 2019.
- [2] Malte D. Luecken, M. Büttner, K. Chaichoompu, A. Danese, M. Interlandi, M. F. Mueller, D. C. Strobl, L. Zappia, M. Dugas, M. Colomé-Tatché, and Fabian J. Theis. Benchmarking atlas-level data integration in single-cell genomics. *Nature Methods*, 19(1):41–50, Jan 2022.
- [3] Jiankun Lyu, Nicholas Kapolka, Ryan Gumpper, Assaf Alon, Liang Wang, Manish K. Jain, Ximena Barros-Álvarez, Kensuke Sakamoto, Yoojoong

Kim, Jeffrey DiBerto, Kuglae Kim, Isabella S. Glenn, Tia A. Tum-
mino, Sijie Huang, John J. Irwin, Olga O. Tarkhanova, Yurii Moroz,
Georgios Skiniotis, Andrew C. Kruse, Brian K. Shoichet, and Bryan L.
Roth. Alphafold2 structures guide prospective ligand discovery. *Science*,
384(6702):eadn6354, 2024.

[4] Ethan Weinberger, Chris Lin, and Su-In Lee. Isolating salient variations of
interest in single-cell data with contrastivevi. *Nature Methods*, 20(9):1336–
1345, Sep 2023.

Responses to comments by Reviewer #3 737

This manuscript introduces TamGen, a transformer-based language model 738
designed for generating target-aware molecules that exhibit desired drug- 739
related characteristics. It employs geometric self-attention to capture the 740
spatial information of proteins and integrates this with the latent vectors of 741
seed molecules, which are learned through a Variational Autoencoder (VAE). 742
This fused information is subsequently fed into the transformer decoder to 743
learn the joint distribution of molecules and proteins. Consequently, Tam- 744
Gen possesses the capability to produce molecules that are both influenced by 745
the target protein and centered around the seed molecules. Furthermore, the 746
authors undertook wet lab experiments, i.e., generating molecules targeting 747
the Tuberculosis ClpP protease and computing their IC₅₀ values. The authors 748
adopted the Design-Refine-Test workflow, enabling iterative improvements of 749
the generated molecules and thereby highlighting TamGen’s effectiveness in 750
generating viable drug candidates. 751

This study is interesting and offers a promising approach to drug design. This 752
paper is clearly written and I can fully follow up their works. However, there 753
are a few concerns that need to be addressed. 754

Reply: We sincerely appreciate the careful review and high recognition of our 755
work. In response to the reviewers’ concerns, we have made several significant 756
improvements: 757

1. Provided a more comprehensive comparison to 3D generation approaches, 758
as suggested by the reviewers in Q1a-3. We also included comparisons with 759
1D SMILES-based models for a balanced evaluation. 760
2. Conducted additional computational experiments to evaluate the effective- 761
ness of the distance-aware attention and data augmentation techniques 762
(Q2), explored the capability of TamGen to maintain its performance on the 763
MRR metric across protein pockets (Q4), and investigated the association 764
between Lipinski’s rule of five and logP (Q5). 765
3. Performed preliminary experiments for comparison with TargetDiff in the 766
real world applications (Q3). We highlighted the iterative improvements 767
through the Design-Refine-Test pipeline. 768
4. Made minor adjustments to improve the visualization and clarity of our 769
figures, specifically enhancing Figure S4 and standardizing the representa- 770
tion of benzene rings in Figure 4a (Q6). 771

With all these updates, we believe that reviewers’ concerns are fully addressed; 772
also thanks to the suggestions, the revision should also enhance the quality 773
and clarity of our manuscript. 774

Question 1: The benchmark is somewhat unbalanced. All methods in the 775
benchmark generate molecules in 3D Euclidean space while TamGen generates 776
molecules in 1D SMILES sequence. Different molecular representations come 777
with their own strengths and weaknesses. 3D representations excel at capturing 778

the interactions between proteins and ligands, whereas 1D representations are
adept at generating molecules with high validity and sensible structures, like
fewer fused rings. Therefore, for a more balanced comparison, the authors
should also consider comparing their methods with 1D SMILES-based models,
as exemplified by references [1-2].

[ref1] Transformer neural network for protein-specific de novo drug gener-
ation as a machine translation problem. Sci Rep. [ref2] AlphaDrug: protein
target specific de novo molecular generation, PNAS Nexus, Volume 1, Issue 4,
September 2022, page227.

**Reply:** Thank you for your valuable feedback and suggestions on incorporating
more 1D SMILES-based models for a balanced comparison. We have carefully
studied the works cited in references [ref1, ref2] and appreciate the suggestion
to compare our methods with these models.

Both [ref1] and AlphaDrug [ref2] have implemented a Transformer encoder
to process protein sequences and a decoder to generate SMILES. Refer-
ence [ref1] trains a conventional Transformer model that encodes a protein
sequence into a series of hidden states before decoding it into SMILES on the
decoder side. The authors of AlphaDrug found that “... *Although the origi-*
*nal transformer has been adopted in [ref1] for the same task, we find that it is*
*not efficient to involve the protein information in molecular generation. The*
*inefficiency is due to the information transfer bottleneck from the encoder top-*
*layer to various levels of decoder layers ...” (words from [ref2]). Therefore,*
AlphaDrug proposes the “Lmsr Transformer” to solve the aforementioned lim-
itation. More importantly, AlphaDrug incorporates a Monte Carlo Tree Search
(MCTS) algorithm, which leverages docking scores to steer the generation of
compounds that exhibit superior docking efficacy.

We have conducted a comparison of TamGen, with AlphaDrug and [ref1]
using the benchmarks provided in the study of AlphaDrug. The results are
summarized in Table R1. We can see that our method outperforms [ref1].
We also have that AlphaDrug, when employing the MCTS-guided generation
strategy, achieves higher docking scores compared to our method. This under-
scores the effectiveness of MCTS in optimizing the properties of generated
molecules. We reason that AlphaDrug’s MCTS algorithm utilizing the dock-
ing score as a direct feedback mechanism in the compound generation process
naturally favors the production of molecules with optimized docking scores,
since the generation strategy is fundamentally guided by this metric. With-
out the MCTS, AlphaDrug’s performance does not surpass our method (see
AlphaDrug w/o MCTS, Table R1).

The MCTS approach presents a promising strategy for molecule genera-
tion in scenarios lacking explicit supervised signals. Looking forward, we plan
to enhance TamGen by integrating MCTS or employing reinforcement learn-
ing techniques to guide molecule generation. These enhancements are expected
to improve various drug properties, including compound stability, synthesiz-
ability, ADME/T properties, and binding affinities. We have included this

comparison in the Discussion section (Line 648 to Line 656) of the paper to better highlight the value of AlphaDrug.

Thank you again for your constructive feedback, which has guided us towards a more balanced and comprehensive evaluation of our approach.

Methods	Docking (\uparrow)	LogP	QED (\uparrow)	SA (\downarrow)
[ref1]	8.5	3.8	0.5	2.8
AlphaDrug w/o MCTS	8.5	4.0	0.5	2.7
AlphaDrug (max)	11.6	5.2	0.4	2.7
TamGen	10.4	3.9	0.5	2.6

Table R1: Comparative Analysis with AlphaDrug. We provided 10 compounds per receptor following the protocols of AlphaDrug and conducted evaluations using AlphaDrug’s metrics. An upward arrow (\uparrow) indicates that a higher value is advantageous, whereas a downward arrow (\downarrow) suggests that a lower value is preferable. “AlphaDrug w/o MCTS” denotes a variant of AlphaDrug that operates without the Monte Carlo Tree Search (MCTS) optimization. The “SA” in this table use a different implementation from the SAS scores reported in the main paper. The results of [ref1], “AlphaDrug w/o MCTS” and “AlphaDrug” are from [ref2].

Question 1a: Besides, more recently published methods [3-6] should be introduced in the main body. [3] Generation of 3D molecules in pockets via a language model. Nat Mach Intell 6, 62–73 (2024). [4] KGDiff: towards explainable target-aware molecule generation with knowledge guidance, Briefings in Bioinformatics, Volume 25, Issue 1, January 2024, bbad435. [5] PocketFlow is a data-and-knowledge-driven structure-based molecular generative model. Nat Mach Intell 6, 326–337 (2024). [6] 3D molecular generative framework for interaction-guided drug design. Nat Commun 15, 2688 (2024).

Reply: We appreciate the reviewer’s suggestion to include more recent work in AI for drug generation. We have thoroughly reviewed the suggested publications and have incorporated brief descriptions to these methodologies in Lines 74-76 of the revised manuscript. Additionally, we have included a paragraph discussing the limitations of TamGen in comparison to 3D generation approaches in the discussion section (Lines 633 to 648). We acknowledge that 3D interactions between compounds and proteins are essential for structure-based drug design. Therefore, while TamGen’s 1D generation methods using SMILES and amino acid sequences offer unique advantages, they may not fully exploit the interactions available in geometric space. In follow-up work, we plan to integrate insights from 3D generation methods, particularly the autoregressive approaches, to enhance TamGen’s capability to utilize 3D interaction patterns. This integration aims to improve compound generation and refinement, potentially yielding compounds with better docking scores and more reasonable structures. Thank you for your valuable suggestions.

**Question 2:** Many deep learning models, such as [7-9], are SE(3)-equivariant
 or SE(3)-invariant with carefully designed constrains on model structures.
 However, TamGen simply learns geometric protein information with data aug-
 mentation to the coordinates and a distance-aware attention, while achieves
 promising performance in wet experiments. It would be better if the authors
 can provide an ablation study to figure out the effectiveness of the two
 techniques.

**Reply:** We thank the reviewer for the thorough reading and valuable sugges-
 tions. To examine the importance of the distance-aware attention and the data
 augmentation technique using in TamGen, we implemented two variants of our
 model. One is without the distance-aware attention (denoted as TamGen w/o
 `dist_attn`), where the equation for distance-aware attention calculation

$$887 \hat{\alpha}_j = \exp\left(-\frac{\|r_i - r_j\|^2}{\tau}\right) (h_i^{(l)\top} W h_j^{(l)}), \quad (1)$$

is replaced by

$$890 \hat{\alpha}_j = h_i^{(l)\top} W h_j^{(l)} \quad (2)$$

Another is without coordinate data augmentation (denoted as TamGen w/o
 `coord_aug`), where the augmentation procedure

$$893 h_i^{(0)} = E_a a_i + E_r \rho \left(r_i - \frac{1}{N} \sum_{j=1}^N r_j \right), \quad (3)$$

is replaced by

$$896 h_i^{(0)} = E_a a_i + E_r \left(r_i - \frac{1}{N} \sum_{j=1}^N r_j \right). \quad (4)$$

The evaluation results are reported in Table R2. The data indicates a notable
 decline in docking scores upon the exclusion of the distance-aware attention
 and 3D coordinate data augmentation. Despite other metrics showing smaller
 changes, this comparison shows the importance of these techniques in captur-
 ing protein information. Given that the protein encoder component of TamGen
 is not pre-trained and suffers from a scarcity of protein data, the incorpora-
 tion of data augmentation and inductive biases (such as the assumption that
 proximate residues are more likely to interact) is advantageous. The perfor-
 mance of TamGen can be further improved by including protein pre-training
 in its future development.

We have included the ablation experiment analysis in the revised
 manuscript (Lines 557 to 564) to better dissect the role of distance-aware
 attention and coordinate data augmentation in TamGen, as you suggested.

**Question 3:** This question serves as a follow-up. Almost all methods in the
 benchmark pointed out that the 3D Euclidean interactions between target pro-
 teins and ligands are crucial for ligands to successfully bind to their respective

	Vina Dock (↓)	QED (↑)	SAS (↑)	Diversity (↑)	LogP ∈ [0, 5]	Lipinski (↑)
TamGen	-7.475	0.559	0.771	0.747	87.9%	98.8%
TamGen w/o <code>dist_attn</code>	-6.943	0.543	0.773	0.761	86.6%	98.4%
TamGen w/o <code>coord_aug</code>	-6.588	0.564	0.774	0.766	75.1%	97.3%

Table R2: Compilation of performance statistics for TamGen and two variants: (i) TamGen w/o `dist_attn` denotes the variant where the distance-aware attention is removed; (ii) TamGen w/o `coord_aug` is the variant where the coordinate augmentation is removed.

target proteins. However, TamGen does not model any above 3D interactions while still achieves good performance in wet experiments. It seems the seeding compounds in Design-Refine-Test pipeline play the key role. If feasible, it would be insightful if the authors could conduct wet lab experiments using molecules generated by other methods that explicitly account for 3D interactions, such as TargetDiff, for comparison. It would be highly significant if the authors could show that explicitly modeling the 3D interactions between proteins and ligands might not be as crucial, provided that the generated compounds can undergo iterative improvements through the Design-Refine-Test pipeline.

Reply: We thank the reviewer for the insightful comment and constructive suggestion.

First, it is important to clarify that the comparison results to 3D generation approaches do not imply that 1D generation is inherently superior. The good performance of TamGen can be attributed to multiple factors, including data curation, model selection and design, as well as pre-training and training strategies. On the other hand, 3D generation approaches are continuously improving and may achieve better results as the quality and quantity of protein-ligand complex 3D structures increase. It is foreseeable that combining TamGen with 3D modeling could enhance performance, given that 3D interactions between compounds and proteins are crucial for drug development. We included this as an opportunity for future development in revised manuscript (Line 633 to Line 647).

Second, as shown in Figures 2a and 2d of the manuscript, TamGen does not exhibit a leading advantage over 3D generation approaches in terms of docking scores. This is reasonable because 3D generation approaches directly exploit the binding pocket 3D structural information and strongly bias towards molecules that fit perfectly into the pockets, therefore yielding better docking scores. However, these molecules might still face challenges in the lack of drug-likeness and low synthetic accessibility, which are both critical for drug development. In contrast, TamGen, as a 1D generation model pre-trained on a large set of high quality chemical compounds, may achieve better results in experimental tests.

Lastly, for a fair comparison, we generated 12,000 compounds for the ClpP target using TargetDiff, as suggested by the reviewer. We docked the compounds generated by TargetDiff, and the results are presented in Figure R3. The docking scores for TargetDiff were inferior to those obtained in both the Design and Refine stages of TamGen, underscoring the advantages of

967 our method for this particular target. It should be noted that the original
TargetDiff method does not support compound refinement.

Furthermore, we analyzed compounds with docking scores better than the
reference compound Bortezomib, as described in the Methods section, and
reported the results in Table R3. The compounds generated by TamGen exhib-
ited significantly better synthetic accessibility and drug-likeness compared to
those generated by TargetDiff. This is in accordance with our analysis on the
drawbacks of current 3D generation approaches.

Since wetlab experiments have high demands on resources and require
longer time, we could only conduct experiments on TamGen generated com-
pounds and analog molecules. We hope that the capability of 3D generation
methods, such as TargetDiff, can be illustrated by solving real world drug dis-
covery problems. In this work, we have to focus on the research centered around
TamGen, and utilize limited resources to validate the generation outcomes.

**Fig. R3:** Comparison of docking scores between TamGen and TargetDiff on
ClpP. Note that the docking scores of TamGen are better than TargetDiff with
p -value < 0.00001 (Mann–Whitney U test).

	QED (\uparrow)	SAS (\uparrow)	LogP $\in [0, 5]$ (\uparrow)	Lipinski (\uparrow)
TamGen	0.445	0.806	0.861	100%
TargetDiff	0.470	0.606	0.636	92.9%

**Table R3:** Comparison between TamGen and TargetDiff on the ClpP target.

**Question 4:** The capability of TamGen to accurately capture the context of
the protein pocket remains unclear. It would be insightful to explore whether
TamGen can maintain its performance on the MRR metric across protein
pockets of varying sizes. Additionally, it's worth investigating if TamGen has
the ability to generate molecules that are sensitive to different protein isoforms.

**Reply:** We highly appreciate the constructive suggestion by the reviewer.
To evaluate the association between protein pocket size and performance, we
categorized the pockets based on the numbers of residues (denoted as $S(\mathbf{x})$)
as follows:

1. If $S(\mathbf{x}) < 40$, pocket \mathbf{x} is defined as a small pocket; 1013
2. If $S(\mathbf{x}) \in [40, 60)$, pocket \mathbf{x} is defined as a medium pocket; 1014
3. If $S(\mathbf{x}) \geq 60$, pocket \mathbf{x} is defined as a large pocket. 1015

We computed the mean reciprocal rank (MRR) values for the generated compounds corresponding to each pocket size category. The results (Figure R4) demonstrate that TamGen consistently performs the best across all three categories defined based on protein pocket sizes. We also included the findings through this analysis in the supplementary notes of the manuscript. In addition, most approaches, except ResGen, tend to maintain consistent performance irrespective of pocket size. 1016

Fig. R4: MRR of the generated compounds of different methods with respect to pocket sizes. 1023

Regarding the challenge of protein isoforms as mentioned by the reviewer, we acknowledge that isoforms pose a significant challenge to drug discovery and design. If an isoform results in substantial changes in the binding pocket, TamGen can be applied to generate compounds tailored to the altered pocket. However, if the differences occur outside the binding pockets or have minimal impact on the pockets, they may not be distinguishable by TamGen. Similarly, current implementation of TamGen is not capable of differentiating between wild-type proteins and proteins with point mutations. 1032

Indeed, as the reviewer suggested, sensitivity to isoforms or mutations is a crucial and appealing capability for drug generation models. We have noted this point as one of the limitations of TamGen in the manuscript (Line 626). In the future, we plan to improve the model to better distinguish between protein isoforms and mutants. 1039

Question 5: Using Mean Reciprocal Rank (MRR) based on both Lipinski's rule of five and logP might introduce a bit of overlap, as Lipinski's rule already takes logP into consideration among its criteria. 1047

Reply: We thank the reviewer for bringing this point to our attention, as we did not explicitly examine the potential overlap in the metrics during model evaluation. It is true that LogP is indeed one of the criteria in Lipinski's rule 1053

of five (Ro5), however, this does not necessarily lead to redundancy in MRR
evaluation, as Ro5 is a categorical metric rather than a continuous one as logP.

To rigorously evaluate the potential overlap between Ro5 and LogP values,
both of which were used in Mean Reciprocal Rank (MRR) calculation in our
study, we undertook a focused analysis by randomly sampling 1 million com-
pounds from the PubChem database and determining their compliance with
Ro5 as well as their LogP values. We then assessed the association between
these two properties.

Given that compliance with Ro5 is a binary outcome, we employed the
Point-biserial correlation coefficient (denoted as r_{pb}), a measure of the strength
of association between a continuous variable (LogP) and a binary variable
(Ro5 compliance), to quantify the relationship. We use the python package
`scipy.stats.pointbiserialr` to calculate the r_{pb} . For the total sample of
1 million compounds, the r_{pb} between Ro5 compliance and LogP values was
found to be 0.146. Further, we filtered compounds whose LogP values fall
within the range of [0, 5] (773.66K in total) and found an r_{pb} of 0.044 between
Ro5 compliance and LogP values for these compounds. Both r_{pb} values suggest
a poor correlation between the two metrics, implying that considering both Ro5
and LogP simultaneously for MRR calculation allows for a more comprehensive
evaluation of the model outputs from different perspectives. Therefore, we
decided to keep the original MRR evaluation score in the revision. Thanks for
bringing this potential issue again.

**Question 6:** minor changes: a) Figure S4 could be enhanced by adjusting the
line width and font size for clearer visualization. b) In Figure 4a, it would be
beneficial to standardize the representation of benzene rings for clarity and
consistency.

**Reply:** Thanks for pointing out the flaws in the illustrations. In our revised
manuscript, we have made the following improvements: a) We have enhanced
Figure S4 by adjusting the line width and font size for clearer visualization.
b) We have standardized the representation of benzene rings in Figure 2d and
Figure 4a for clarity and consistency. Thank you for your valuable suggestions
to improve the quality of our figures.

Responses to comments by Reviewer #4 1105

The authors introduce TamGen, a target-aware language model for generating potential drug-like molecules. They demonstrate the effectivity of their model by comparison with others similar, and implement a drug-design pipeline using their model where they in the end obtain a few candidates with experimentally validated outstanding properties. 1106

The work offers a novel approach to generating molecules in a targeted manner, and provide evidence of its effectiveness. Some of the claims are potentially not well supported by evidence, and would require either further experiments or rephrasing. 1112

Reply: We express our sincere gratitude for your professional review and valuable feedback on our article. We highly appreciate your constructive suggestions and the identification of areas requiring further experimentation and clarity. In response to your comments, we have made the following adjustments, which enhance the quality of our manuscript after revision: 1116

1. Conducted additional experiments, including the randomization of (pocket, ligand) pairs during training, to demonstrate that TamGen effectively leverages information from these pairs rather than relying on spurious correlations. 1121
2. Analyzed the difference between the Transformer component of the GPT model and RNN, and assessed the performance of the latter architecture. 1125
3. Investigated the binding mechanisms of some generated and experimentally validated compounds. We included visualizations of attention maps to provide insights into how the model identifies these connections, potentially explaining the better docking scores observed. 1127
4. Addressed typographical and illustration inconsistencies, specifically correcting the typo on line 473 (previously line 454) (“urine” to “urea”) and standardizing the representation of benzene rings in Figures 2d and 4a. 1131

We believe these revisions and additional experiments strengthen the evidence supporting the effectiveness of TamGen and enhance the overall quality of the manuscript. 1135

Question 1: randomize the (pocket, ligand) pairs during training, and reproduce Fig2b. That way you’ll prove that the model is actually leveraging information from the (pocket, ligand) pairs instead of some spurious correlation. 1139

Reply: This is an excellent suggestion to further evaluate the performance of TamGen. We sincerely appreciate the reviewer’s constructive comments regarding the randomization experiment. To examine whether TamGen has captured the relationship between the pocket and the ligand, we randomized the (pocket, ligand) pairs and re-trained a model TamGen-r (with “r” stands for random). Specifically, given the training dataset $\mathcal{D} = \{(\mathbf{x}_i, \mathbf{y}_i)\}_{i=1}^N$, we shuffled the indices $I = \{1, 2, \dots, N\}$ using the python code `random.shuffle(I)` 1143

and obtained the shuffled indices $\{\zeta(1), \zeta(2), \dots, \zeta(N)\}$. Consequently, we
 obtained a randomized dataset $\tilde{\mathcal{D}} = \{(\mathbf{x}_{\zeta(i)}, \mathbf{y}_i)\}_{i=1}^N$. We then followed the same
 training and inference procedure as the standard TamGen for TamGen-r. The
 results, summarized in Table R4, show that the docking scores for TamGen-
 r are significantly worse in comparison to those of TamGen. These results
 demonstrate that TamGen effectively captures the relationship between the
 pocket and the ligand and leverages such information for molecular generation.

	Vina Dock (\downarrow)	QED (\uparrow)	SAS (\uparrow)	Diversity (\uparrow)	LogP \in [0, 5] (\uparrow)	Lipinski (\uparrow)
TamGen	-7.475	0.559	0.771	0.747	87.9%	98.8%
TamGen-r	-6.842	0.522	0.762	0.731	91.1%	100%

**Table R4:** Comparison between TamGen and its randomized variant
 TamGen-r. The variant TamGen-r is trained on the dataset where a pocket
 information is randomly shuffled.

 While the performance gap between the two models might be expected, we
 argue that: 1. The absolute value of the docking score may not be highly infor-
 mative, as it is a virtual estimation instead of wetlab experimental results. The
 trend, however, indicates the goodness of the interaction between pockets and
 compounds. 2. Given that we are providing valid protein pockets for drug dis-
 covery, these pockets may share certain similarities, resulting in borderline level
 docking scores predicted by the TamGen-r model. 3. Other metrics describe
 the properties of the generated compound alone and may not be significantly
 affected by the randomization experiment.

**Fig. R5:** The relation between synthetic accessibility scores (SAS)
 and docking scores of TamGen and TamGen-r..

Upon reproducing Fig. 2b in the original manuscript, as suggested by the reviewer, we found that TamGen showed a clear advantage in docking scores when the synthetic ability score (SAS) was below 0.8. However, for compounds with an SAS above 0.8, TamGen-r surpassed TamGen (Fig. R5). One possible explanation is that the pre-trained compound decoder can generate chemically sounding molecules regardless of the conditions provided by the encoder. We appreciate the reviewer for bringing this phenomenon to our attention, and it will be interesting to further analyze the molecules generated by the two models in detail in the future. Overall, these results still demonstrate that the original TamGen generates compounds with better docking scores for target molecules.

We have included this experiment in the Ablation section (Lines 550 to 556) and Methods section (Lines 943 to 951) of the manuscript to demonstrate that TamGen captures the pocket-ligand information, as suggested by the reviewer.

Question 2: The authors claim that the GPT model can leverage few data efficiently. Could you assess the effect of the model type to support this claim? e.g. replace the GPT part by an RNN.

Reply: We appreciate the reviewer’s insightful question regarding the efficiency of GPT models in leveraging limited data, and the potential impact of model type on this capability. To clarify, our statement in the manuscript: “The pre-training of the compound decoder using chemical compound information, similar to GPT models, is a core component of TamGen” aims to highlight the importance of pre-training rather than to claim that GPT models inherently leverage small datasets efficiently. We recognize that pre-training, coupled with model and data scaling, is pivotal for the success of models like GPT. Hence, we have revised the manuscript to avoid any misconceptions regarding this point. Moreover, we would like to address the reasons for selecting Transformers over RNNs in our work:

1. Parallelization:

- Transformers: These models process entire sequences in parallel, significantly accelerating training, particularly with GPU support.
- RNNs: They process sequences step-by-step, hindering parallelization and resulting in slower training times.

2. Handling Long-Range Dependencies:

- Transformers: The self-attention mechanism in Transformers effectively captures long-range dependencies, allowing each token to attend to any other token in the sequence.
- RNNs: Due to their sequential nature, RNNs struggle with long-range dependencies and suffer from the vanishing gradient problem.

3. Scalability:

- • Transformers: These models are highly scalable and demonstrate
 improved performance when scaled up (e.g., models like BERT and
 GPT-3).
 • RNNs: Scaling is more challenging due to sequential processing, which
 limits the benefits of larger models.

4. Pre-training and Transfer Learning:

- • Transformers: They have proven extremely successful in pre-training
 and fine-tuning paradigms, achieving state-of-the-art performance in
 numerous NLP tasks, exemplified by models like GPT-4 and LLAMA.
 • RNNs: While they can be pre-trained, they have not reached the same
 level of success and adoption as Transformers.

In light of these advantages, we believe that Transformer models represent
 the optimal choice for our application. Nevertheless, to thoroughly evaluate
 the impact of model type, we conducted an experiment where we replaced
 the Transformer-based decoder in TamGen with LSTM layers (a widely used
 RNN architecture). We experimented with various numbers of decoder lay-
 ers ($L = 1, 2, 4$) and observed that a single LSTM layer ($L = 1$) performed
 best in our implementation. As indicated in the table below (Table R5), the
 Transformer-based model significantly outperforms the RNN-based model in
 terms of docking score with p -value $< 10^{-10}$.

	Vina Dock (↓)	QED (↑)	SAS (↑)	Diversity (↑)	LogP ∈ [0, 5]	Lipinski (↑)
TamGen	-7.475	0.559	0.771	0.747	87.9%	98.8%
LSTM	-6.364	0.607	0.791	0.725	96.7%	99.9%

**Question 3:** Other experiments, potentially future work: The authors inves-
 tigate the binding mechanisms of some of the generated and experimentally
 validated compounds. Could you prove the model is somehow aware of these
 connections and learns to find them? Or what is the explanation for the better
 docking scores overall? Maybe visualizing attention maps could help this.

**Reply:** It is indeed useful to conduct an investigation of binding mecha-
 nisms and the potential use of attention maps for interpreting our model's
 performance.

As demonstrated in Figure 2, methods, especially those follow 3D gener-
 ation approaches, are capable of generating molecules directly within the 3D
 protein pocket, and they may yield compounds with superior docking scores
 compared to 1D generation methods that inherently face challenges in opti-
 mizing docking interactions. We have discussed these limitations and potential
 solutions in the updated Discussion section (highlighted in red).

While attention weights in Transformers have been widely utilized for
 interpretability in domains such as natural language processing and protein

structure modeling, our experience suggests that the complexity of the model, due to the number of layers and attention heads, necessitates careful interpretation. Attention weights can be influenced by normalization and activation layers, and thus may provide only indirect clues about the model's design principles.

Fig. R6: Attention map and protein-ligand interactions. (a) Proposed binding modes of Syn-A003-01 against ClpP. (b) Numbering of the atoms. (c) attention map. Red boxes indicate agreement between proposed binding modes and the attention map.

Following the reviewer's suggestion, we conducted an attention map analysis for the compound Syn-A003-01, which TamGen generated for Mtb ClpP. We averaged the attention weights from last three layers and made the following observations:

- we found that amino acids located at different sequential positions within the pocket were grouped into the same cluster (Fig. R6c). This indicates that TamGen learns information beyond the sequential order of the protein, suggesting an understanding of the spatial arrangement within

1335 the pocket. Clustering analysis was done with hierarchical clustering
(“scipy.cluster.hierarchy.linkage”).

• The clustering analysis also revealed that TamGen grouped three key oxygen
atoms (O4, O8, and O20), which are crucial for protein-ligand interactions,
together (Fig. R6c). Similarly, the warhead features of the compound (Cl29,
C33, F35, F38, F40), were closely linked. This clustering result suggests that
TamGen captures the importance of specific atoms in the binding process
(Fig. R6c).

• Interestingly, TamGen identified several interactions that were corroborated
by docking experiments, as shown in Fig. R6a-b. For instance, interactions
such as L126-O20, N154-O4/O8, H123-O4/O8, and I71-N22 were captured
by the model, demonstrating its capability to recognize critical binding
interactions.

These findings, while preliminary, suggest that TamGen not only generates
compounds with favorable docking scores but also captures essential binding
interactions within the protein-ligand complex. We believe that further explo-
ration could provide deeper insights into the model’s learning process and
contribute to future improvements.

Thank you again for your valuable feedback.

**Question 4:** Apart from that, fix this typo in the manuscript: (1) line 454:
change “urine” to “urea”; (2) In Figures 2d and 4a, there is an inconsistent use
of kekulized structures along with resonance structures (explicit double bonds
in aromatic ring). I would advice to select one and stick to it for publication.

**Reply:** We would like to thank the reviewer for pointing out these issues. We
have made the following corrections: 1. Fixed the typo on Line 473 (previously
line 454) by changing “urine” to “urea”. 2. Unified the illustration of benzene
rings in Figures 2d and 4a for consistent representation, as suggested. Thank
you for helping us improve the quality of the work.

Reviewer #2 (Remarks to the Author):

I believe that the limitations have been clearly articulated.

All of our points have been addressed appropriately, and I see no further need for additional comments. While the method is still limited in some applications, I acknowledge that this has been clearly communicated in the revision.

Reviewer #3 (Remarks to the Author):

Upon reviewing the author's rebuttal to my previous comments, I find that the issues I raised have been adequately addressed. I believe that the quality of the article now meets the publication standards.

I recommend that the paper be accepted for publication.

Comment on Reviewer #1's comments:

I have carefully read the comments and questions raised by the reviewer, and the response written by the authors. The authors have addressed the concerns very well.

Reviewer #4 (Remarks to the Author):

The authors propose method based on language models for drug-like molecule generation that is aware of the protein target. the method achieves competitive results against other target-aware molecular generation algorithms and is consistently top-ranked in the evaluation metrics.

The authors further evaluate the proposed model in a case-study (design of inhibitors for Tuberculosis ClpP protease) where the model is used for designin, screening and suggesting modifications to compounds. Selected compounds are synthesized and evaluated for the target property. This led to multiple compounds with $IC_{50} < 20 \mu M$, of which one has $IC_{50} = 1.9 \mu M$. The authors further perform ablation studies to validate the contributions of each component and to show that the model effectively uses the structured information given during training time.

The proposed model is original in relation to other models developed for the same purpose, as it uses as language model to generate molecules (as SMILES string) while incorporating information from the target pocket and potentially a seed compound, while other cited works focus on 3D molecular generation, generation based on reinforcement learning, and other methods.

A potentially similar model is reported in Gao, Zhangyang et al. "PrefixMol: Target- and Chemistry-aware Molecule Design via Prefix Embedding." ArXiv abs/2302.07120 (2023) <https://arxiv.org/abs/2302.07120>. This model was reported in February 2023, and it reports a GPT-based model that can be prefixed (conditioned) with multiple relevant information including the 3D pocket. This information is prefixed in the form of modality-specific vector embeddings. Nevertheless, the cited work doesn't provide as much insight as the draft in review

does, and apparently the proposed model does not perform as good. Still, I suggest the authors give this paper a revision and cite it, clarifying differences between the two models.

The draft is additionally original in that it carries out a drug-design pipeline for a specific usecase, including synthesis and experimental evaluation in the end which gives a trustworthy demonstration of the model's capabilities.

Reviewer #4 (Remarks on code availability):

The code looks complete enough to reproduce the results, however I couldn't due mainly to:

- The link for downloading the model is not working, seems that the file doesn't exist, and
- The README doesn't provide enough details for reproducing the dataset generation and training of the model.

For example, the first command provided in the README

```
python scripts/build_data/prepare_pdb_ids.py ${PDB_ID_LIST} ${DATASET_NAME} -o  
${OUTPUT_PATH} -t ${threshold}
```

requires crucially the inputs: `PDB_ID_LIST` and `threshold`. Although it is explained that PDB_ID_LIST is a file containing a list of PDB IDs, I find it relevant for reproducibility that the exact file used here is released as part of the code, or made available elsewhere, as this file is crucial for the results downstream. Additionally, the parameter `threshold` is given without explanation and it is not clear what it refers to exactly. I couldn't find reference to its meaning in the code, nor the draft or the SI.

Regarding the data, in the draft the authors reference [1] and say that they use the same dataset. Inspection of that paper leads to

<https://github.com/pengxingang/Pocket2Mol/tree/main>, which contains a README that leads directly to a link containing the dataset used. The authors could implement a README similar to that, that points directly to the links to the relevant datasets.

[1] Peng, X., Luo, S., Guan, J., Xie, Q., Peng, J., Ma, J.: Pocket2mol: Efficient molecular sampling based on 3d protein pockets. International Conference on Machine Learning (2022)"

Responses to comments by Reviewer #4

General comments 1: The authors propose method based on language models for drug-like molecule generation that is aware of the protein target. the method achieves competitive results against other target-aware molecular generation algorithms and is consistently top-ranked in the evaluation metrics. The authors further evaluate the proposed model in a case-study (design of inhibitors for Tuberculosis ClpP protease) where the model is used for designing, screening and suggesting modifications to compounds. Selected compounds are synthesized and evaluated for the target property. This led to multiple compounds with $IC_{50} < 20 \mu M$, of which one has $IC_{50} = 1.9 \mu M$. The authors further perform ablation studies to validate the contributions of each component and to show that the model effectively uses the structured information given during training time. The proposed model is original in relation to other models developed for the same purpose, as it uses as language model to generate molecules (as SMILES string) while incorporating information from the target pocket and potentially a seed compound, while other cited works focus on 3D molecular generation, generation based on reinforcement learning, and other methods.

Reply: Thanks for your positive feedback and the precise summary of the present work. Please find our further refinement and response below.

Question 1: A potentially similar model is reported in Gao, Zhangyang et al. "PrefixMol: Target- and Chemistry-aware Molecule Design via Prefix Embedding." ArXiv abs/2302.07120 (2023) <https://arxiv.org/abs/2302.07120>. This model was reported in February 2023, and it reports a GPT-based model that can be prefixed (conditioned) with multiple relevant information including the 3D pocket. This information is prefixed in the form of modality-specific vector embeddings. Nevertheless, the cited work doesn't provide as much insight as the draft in review does, and apparently the proposed model does not perform as good. Still, I suggest the authors give this paper a revision and cite it, clarifying differences between the two models. The draft is additionally original in that it carries out a drug-design pipeline for a specific usecase, including synthesis and experimental evaluation in the end which gives a trustworthy demonstration of the model's capabilities.

Reply: Thanks for pointing to PrefixMol, which is indeed very relevant to TamGen. We revised the manuscript to provide a more comprehensive review of relevant work.

Both TamGen and PrefixMol are designed to encode the protein pocket information and decode 1D molecular SMILES. Beyond leveraging this pocket information, the properties of the compounds, such as VINA, QED, SA, LogP, and Lipinski's rules, are integrated into the PrefixMol. Such implementation enables property driven molecular generation, i.e., by specifying single or multiple input properties, one can obtain tailored outputs, making this process

047 highly adaptable. This methodology is promising and holds potential synergies
048 with our work, which we plan to explore in future integrations.

049 In the submitted work, we propose a “Design-Refine-Test” pipeline with
050 TamGen as the core component. TamGen is trained under the variational auto-
051 encoder framework, which makes it easy to implement the “Refine” stage by
052 the contextual encoder (right panel, Fig. 1(b)). PrefixMol opts for a direct
053 encoding strategy, and additional efforts are required to implement molecular
054 refinement. We believe that each model contributes valuable insights into the
055 development of computational tools for molecular design, and we acknowledge
056 the potential for diverse strategies to advance the field in complementary ways.

057 We revise the paper accordingly, in the discussion section (Lines 657
058 to 660): A recent work, PrefixMol [60], also aims to generate
059 SMILES based on both the pocket information and the compound
060 properties, such as QED and SA scores. Overall, the property
061 guided generation points to an important direction for future
062 development.

063

064 **Question 2:** The code looks complete enough to reproduce the results,
065 however I couldn’t due mainly to.

066

- 067 1. The link for downloading the model is not working, seems that the file
068 doesn’t exist, and
- 069 2. The README doesn’t provide enough details for reproducing the dataset
070 generation and training of the model.

071 **Reply:** Thanks for raising the issue in code accessing and the documentation.
072 We have updated the code sharing link and double checked the accessibil-
073 ity. The README document is enhanced with more detailed instructions for
074 testing and applications of TamGen.

075 Our code is at <https://github.com/SigmaGenX/TamGen>.

076 We update the link to download the model, which is at https://1drv.ms/f/s!AoD_zA0b9YjocKiSXTysU5Cwyn4?e=c1QlfA. You can find the model in
077 the “ckpt” folder of the provided link.
078

079 We offer an extensive and detailed README that guides users through
080 various stages such as data preparation, building a customized dataset, model
081 training, etc. For detailed instructions on replicating the dataset generation
082 process, please refer to the “data” directory of the repository. For model train-
083 ing procedures, refer to the [https://github.com/SigmaGenX/TamGen?tab=](https://github.com/SigmaGenX/TamGen?tab=readme-ov-file#training)
084 [readme-ov-file#training](https://github.com/SigmaGenX/TamGen?tab=readme-ov-file#training), where we have documented the default parameters
085 utilized during our model’s training phase.

086

087 **Question 3:** For example, the first command provided in the README
088 `python scripts/build_data/prepare_pdb_ids.py ${PDB_ID_LIST} \
089 ${DATASET_NAME} -o ${OUTPUT_PATH} -t ${threshold}`

090

091

092

requires crucially the inputs: `PDB_ID_LIST` and `threshold`. Although it is explained that `PDB_ID_LIST` is a file containing a list of PDB IDs, I find it relevant for reproducibility that the exact file used here is released as part of the code, or made available elsewhere, as this file is crucial for the results downstream. Additionally, the parameter `threshold` is given without explanation and it is not clear what it refers to exactly. I couldn't find reference to its meaning in the code, nor the draft or the SI.

Reply: Sorry for the confusion and thanks for pointing out this issue. The code in question is designed to create a customized dataset for training and inference. We have included a comprehensive example in the `customized_example` directory within the repository to demonstrate the process of constructing a customized dataset and performing inference on it. In this example, the `PDB_ID_LIST` is available at https://github.com/SigmaGenX/TamGen/blob/main/customized_example/reference_results/7vh8_out.csv.

We also provide the PDB id list we used to train that model that is used to generate compounds for ClpP, which is in `data/pdb_id.csv` of https://1drv.ms/f/s!AoD_zA0b9YjocKiSXTysU5Cwyn4?e=c1QlfA

The parameter `threshold` is used to determine the pocket region.

1. If one uses `scripts/build_data/prepare_pdb_ids_center.py` or `scripts/build_data/prepare_pdb_ids_center_scaffold.py` to determine the pocket region, then `$threshold` denotes radius of the pocket region.
2. If one uses `scripts/build_data/prepare_pdb_ids.py` to determine the pocket region, then the pocket region is defined as follows: A residue r is considered part of the pocket region, if any atom in r lies within `$threshold` angstroms of a ligand atom. For a given `pdb_id`, its associated ligands can be found in the `database/PdbCCD` folder of our repo.

The above information is updated in <https://github.com/SigmaGenX/TamGen/tree/main?tab=readme-ov-file#build-customized-dataset>

Question 4: Regarding the data, in the draft the authors reference [1] and say that they use the same dataset. Inspection of that paper leads to <https://github.com/pengxingang/Pocket2Mol/tree/main>, which contains a README that leads directly to a link containing the dataset used. The authors could implement a README similar to that, that points directly to the links to the relevant datasets.

[1] Peng, X., Luo, S., Guan, J., Xie, Q., Peng, J., Ma, J.: Pocket2mol: Efficient molecular sampling based on 3d protein pockets. International Conference on Machine Learning (2022)”

Reply: We provide a step-by-step README in <https://github.com/SigmaGenX/TamGen/tree/main/data> to process the data. We also upload a copy of the data in the `data/crossdocked_data.zip` of https://1drv.ms/f/s!AoD_zA0b9YjocKiSXTysU5Cwyn4?e=c1QlfA

Reviewer #4 (Remarks to the Author):

I thank the authors for their effort. I believe my comments have been adequately addressed and I think the draft now meets publication standards. I recommend acceptance and publication.

Reviewer #4 (Remarks on code availability):

All my concerns regarding the code have been addressed including the update of links to model checkpoints and input files.

Dear editors and reviewers,

We have addressed all review comments from the previous round, ensuring there are no outstanding questions from the reviewers.

Our primary focus is on the questions in the author checklist, which you can refer to for our responses.

Regards,
Yingc